# Interactive ocean bathymetry and coastlines for simulating the last deglaciation with the Max Planck Institute Earth System Model (MPI-ESM-v1.2)

Virna Loana Meccia[1] and Uwe Mikolajewicz[1]

[1]Max Planck Institute for Meteorology, Bundesstraße 53, 20146 Hamburg, Germany

*Correspondence to*: Virna Loana Meccia (virna.meccia@mpimet.mpg.de)

**Abstract.** As ice sheets grow or decay, the net flux of freshwater into the ocean changes and the bedrock adjusts due to isostatic adjustments, leading to variations in the bottom topography and the oceanic boundaries. This process was particularly intense during the last deglaciation due to the high rates of ice-sheet melting. It is, therefore, necessary to consider transient ocean bathymetry and coastlines when attempting to simulate the last deglaciation with Earth System Models (ESMs). However, in most standard ESMs the land-sea mask is fixed throughout simulations because the generation of a new ocean model bathymetry implies several levels of manual corrections, a procedure that is hardly doable very often for long runs. This is one of the main technical problems towards simulating a complete glacial cycle with general circulation models.

For the first time, we present a tool allowing for an automatic computation of bathymetry and land-sea mask changes in the Max Planck Institute Earth System Model (MPI-ESM). The algorithms developed in this paper can easily be adapted to any free-surface ocean model that uses Arakawa-C grid in the horizontal and z-grid in the vertical including partial bottom cells. The strategy applied is described in detail and the algorithms are tested in a long-term simulation demonstrating the reliable behaviour. Our approach guarantees the conservation of mass and tracers at global and regional scales, that is, changes in a single grid point are only propagated regionally. The procedures presented here are an important contribution to the development of a fully coupled ice sheet-solid earth-climate model system with time-varying topography and will allow for transient simulations of the last deglaciation considering interactive bathymetry and land-sea mask.

## 1 Introduction

During the last deglaciation, the Earth transitioned from the last glacial to the present interglacial climate, experiencing a series of abrupt changes on decadal to millennium timescales. The culmination of the last glacial cycle is denoted by the Last Glacial Maximum (LGM; ca. 21 thousand years before present (ka BP)) characterized by large ice sheets, cold oceans and low greenhouse gas concentrations (Braconnot et al., 2007a; 2007b). During the LGM, vast ice sheets covered large regions of the Northern Hemisphere (e.g. Boulton et al., 2001; Dyke et al., 2002; Svendsen et al., 2004; Tarasov et al., 2012; Peltier et al., 2015), whereas the Antarctic Ice Sheet expanded to the edge of the continental shelf (Argus et al., 2014; Briggs et al.,

2014; Lambeck et al., 2014 and references therein). The global annual mean surface temperature is estimated to have been 4.0 ± 0.8 degrees colder than today (Annan and Hargreaves, 2013); it started to increase towards the present value around 19 ka BP (Jouzel et al., 2007; Buizert et al., 2014). The reason of such a rise is attributed to an increase in the summer insolation at the northern hemisphere high latitudes and in the global atmospheric greenhouse gas concentrations (Berger, 1978; Loulergue et al., 2008; Marcott et al., 2014; Bereiter et al., 2015).

Nowadays, some complex Earth System Models (ESMs) can be run for multiple millennia becoming powerful tools for investigating the mechanisms underlying these climate change events. Especially, transient simulations of the last deglaciation might be valuable for examining the behaviour of non-stationary climate systems and the ice-ocean-atmosphere interactions. Since 1990's, the Paleoclimate Modeling Intercomparison Project (PMIP) aims at evaluating the performance of state-of-the-art climate models in simulating well-documented climates outside the range of present variability (Kageyama et
al., 2018). Recently, PMIP has established the Last Deglaciation Working Group to coordinate the efforts to run transient simulations of the last 21 ka BP (Ivanovic et al., 2016). According to the authors, one aspect to be considered is the varying orography, ocean bathymetry, and land-sea mask. This is because changes in the ice sheets during the deglaciation affected continental topography and ocean bathymetry, which in turn moved the coastal boundaries. Differences in ocean bathymetry and land-sea mask between present-day conditions and 21 ka BP calculated from the ICE-6G_C ice-sheet reconstructions
(Argus et al., 2014; Peltier et al., 2015) are plotted in Fig. 1. In general, the topography of the NH ice sheets does not vary substantially between different reconstructions whereas uncertainties show larger for Antarctica (Abe-Ouchi et al., 2015). Values up to 125 meters in ocean depth variations (Fig. 1a) are estimated, representing deepening of the ocean with time. The largest changes in the oceanic boundaries occurred in the northern hemisphere where the extensive areas covered by ice sheets during the LGM were flooded due to the ice melting (blue areas in Fig. 1b). It is important, therefore, to consider
these changes when attempting to simulate the last deglaciation, for example by including a varying ocean surface area and volume.

There are many examples in the literature of the use of climate models to study the last deglaciation. Some works focus on the timing of the deglaciation (Liu et al., 2009; Menviel et al., 2011; Roche et al., 2011) and other studies address a particular component of the climate system. Many authors have investigated the effects of the glacial forcing on the atmosphere
(Justino et al., 2005; Pausata et al., 2011; Otto-Bliesner et al., 2014) or on the ocean thermohaline circulation (Kim, 2004; Brady et al., 2013; Klockmann et al., 2016). Moreover, some research was carried out by using comprehensive climate and ice-sheet models (Abe-Ouchi et al, 2013) or climate models interactively coupled with a dynamic ice-sheet model for studying the last glacial-interglacial cycles (Bonelli et al., 2009; Heinemann et al., 2014; Ganopolski et al., 2016) and more specifically, the LGM (Ziemen et al., 2014). Still, in standard ESMs, land-sea mask is traditionally treated as fixed. There are
studies that consider a time-varying orography in coupled atmosphere-ocean-ice sheet models (e.g. Ridley et al., 2005, Mikolajewicz et al., 2007a and 2007b, Ziemen et al., 2014) but an interactive ocean bathymetry and coastlines for an ocean model have not been done yet. Liu et al. (2009) performed a transient simulation with the NCAR CCSM3 and they manually

updated the ice-sheet topography about every 1000 years and the meltwater scheme at longer and irregular intervals over the deglacial period. In the PMIP4 last deglaciation Core experiment design, the bathymetry and land-sea mask are considered boundary conditions that cannot evolve automatically in the model. Thus, the decision of how often to make manual updates was left to the expert (Ivanovic et al., 2016). However, by varying the bathymetry in small steps, the artificial signals produced by changes in the ocean configuration might be reduced yielding to a more realistic representation of the ocean circulation and its interaction with the other climate components during the last deglaciation. It might be possible to produce a set of topographies in advance for several time slices with the aim of performing simulations with prescribed topography and ice sheets. However, when approaching the problem of simulations of the deglaciation with a fully interactive ice sheet-solid earth-climate model where the topographies and their changes are a prognostic variable, the need for an automatic procedure becomes more urgent. In such a model system, similar problems would also occur in long-term simulations of anthropogenic climate change.

Our long-term goal in the context of the project "From the Last Interglacial to the Anthropocene: Modeling a Complete Glacial Cycle – (PalMod)", is to simulate the last termination with a coupled ice sheet-solid earth-climate model with interactive coastlines and topography forced only with solar insolation and greenhouse gases concentration. The planned model set up consists of the MPI-ESM (Giorgetta et al., 2013) coupled to the modified Parallel Ice Sheet Model (mPISM) and the Viscoelastic Lithosphere and Mantle model (VILMA). Hence, an automatic procedure to calculate a new set of masks, orographies and bathymetries together with adequate algorithms to transform the restart files that allows for the conservation of various properties is essential. The effect of orography changes on terrestrial runoff using a hydrological discharge (HD) model is treated as in Riddick et al. (2018). In this paper, we focus on interactive changes in ocean bottom topography and land-sea mask in the ocean component of MPI-ESM.

Dealing with interactive bathymetry and land-sea mask in ocean models is challenging from a technical point of view but is necessary for adequately simulating the last deglaciation with GCMs. Indeed, changes in bottom topography and oceanic boundaries during deglaciation were particularly large in the northern hemisphere (Fig. 1) where North Atlantic Deep Water formation takes place. Hence, they should be taken into consideration to get an appropriate representation of the deep ocean circulation during the last deglaciation. However, the generation of an ocean bathymetry to run a model usually implies several checks and manual corrections. This is a necessary step in order to, for example, avoid isolated wet points or inland lakes in the ocean domain. Additionally, it is crucial to look into details, whether passages, islands and peninsulas are correctly represented. If necessary, they should be modified by connecting artificial lakes to the open ocean or connecting artificial islands to the mainland. Repeating this manual procedure continuously is not feasible in very long-term simulations. Hence, to consider the effects of changing bottom topography and coastlines, it is essential to design an automatic procedure. Following this purpose, we present for the first time a tool allowing for the automatic computation of bathymetry and land-sea mask changes in the Max Planck Institute Ocean Model (MPIOM). In our approach, we account for the conservation of mass and water properties at both, global and regional scales, thus avoiding artificial long-distance propagation of signals.

The current version is tailored to a coarse resolution setup of MPIOM, but the extension to other setups is rather straightforward.

The paper is organized as follows. The ocean model specifications to apply our algorithms is discussed in Sect. 2. The methodology for automatically changing the bathymetry and the land-sea mask in MPIOM is detailed in Sect. 3. It contains
the strategy to generate a coarse topography from a high-resolution data (Sect. 3.1), the methodology to gradually change the bathymetry and land-sea mask along the last deglaciation (Sect. 3.2), the solution for adjusting the ocean bottom floor in order to match changes in ocean volume and freshwater fluxes into the ocean (Sect. 3.3), and the approach to modify the restart file with the aim of conserving mass and tracers when the ocean configuration changes (Sect. 3.4). The behaviour of the algorithms within a transient simulation with MPI-ESM-1.2.00p4 is evaluated in Sect. 4. Finally, the strengths and
limitations of our approach and its applicability are discussed in Sect. 5.

## 2 Ocean model requirements

The algorithms presented in this paper are tailored for the coarse resolution setup of MPIOM but should be easily transferable to other model resolutions or other ocean models having similar assumptions and approximations. MPIOM is a
free-surface ocean general circulation model with the hydrostatic and Boussinesq approximations and incompressibility is assumed. It solves the primitive equations on an Arakawa-C grid in the horizontal and a z-grid in the vertical (Maier-Reimer 1997). For freshwater, a mass-flux boundary condition is implemented. A detailed description of the model equations and its physical parametrizations is given in Marsland et al. (2003) while its performance as the ocean component of the MPI-ESM is evaluated by Jungclaus et al. (2013). MPIOM includes an embedded dynamic/thermodynamic sea-ice model (Notz et al.,
2013) with a viscous-plastic rheology following Hibler (1979). Sea ice is floating in the ocean. Ice shelves are not included. In this paper, we use the MPIOM coarse resolution configuration with a curvilinear orthogonal grid (GR30) and two poles (Haak et al., 2003), over Greenland and Antarctica. We decide to use the coarse configuration to reduce the computational time, but the algorithms presented in this paper can easily be adapted to higher resolution grids. In the vertical, the model has 40 unevenly spaced levels, ranging from 15 meters near the surface to several hundred meters in the deep ocean. Vertical
discretization includes partial vertical grid cells. Therefore, at each horizontal grid point, the deepest wet cell has a thickness that is adjusted to resolve the discretized bathymetry. On the other hand, the surface layer thickness is also adjusted to account for the sea surface elevation and the sea ice/snow where appropriate.

## 3 Methodology

As a starting point, we build the tool for automatically dealing with changes in bathymetry and land-sea mask for the coarse resolution configuration MPI-ESM-CR. This configuration is used for paleoclimate applications and corresponds to

approximately 3 degrees horizontal resolution and 40 vertical levels (denoted as GR30) in the ocean component MPIOM. Despite the relatively coarse resolution, it is important to carefully consider the bathymetric details to avoid an unrealistic representation of the ocean floor. We pay particular attention to three aspects. First, we consider the land-sea mask with emphasis on the opening or closure of key straits and channels. Second, the bathymetry of the same straits is treated in order to provide an adequate through-flow-depth (TFD). We assume here that to appropriately simulate the ocean circulation it is more important to conserve the correct TFD rather than the through-flow area. Having an adequate depth of the outflow (e.g. in the Strait of Gibraltar) yields a better representation of the water properties in our coarse resolution model. Finally, we check for the presence of lakes in the GR30 bathymetry; the Caspian Sea and the Black Sea (under LGM condition, for example) are the only cases that are permitted. Because we are dealing with an ocean model, we are interested in lakes that are connected to the ocean, that is the Black Sea. However, we include the Caspian Sea in our calculations because of its potential impact on the climate of Central Asia. Solving the SST of the Caspian Sea, which is much larger than other minor lakes, might be important for coupled climate simulations. All other lakes need to be removed from the ocean domain either by connecting them to the open ocean or by considering them as land. The atmospheric model component allows accounting for lakes on land (only the thermal component). In the framework of our model system, the adequate place to calculate water storage in lakes would be the hydrological discharge model, which is part of the land module.

Our aim is to perform the above-mentioned controls in an automatic way and, therefore, as a starting point, a high resolution (HR) bathymetry is necessary to obtain information on the small features. In what follows, we call HR a 10' × 10' gridded dataset. We use a remapped RTopo-2 bedrock topography for present-day conditions. RTopo-2 (Schaffer et al., 2016) is a compilation of consistent maps of global ocean bathymetry, upper and lower ice surface topographies and global surface height on a spherical grid with 0.5' horizontal resolution. The RTopo-2 topography was remapped to a 10' × 10' regular grid. Remapping the data, in this case, results from a compromise between the horizontal resolution and the computation time for performing the algorithms, especially when reducing the resolution. Because the bathymetry and land-sea mask need to be adjusted several thousand times during deglaciation, it is crucial to construct a fast tool. The aim here is to speed up the computation by remapping RTopo-2 data without losing the general features. On the other hand, our next goal is to couple an ice-sheet and a solid earth model to MPI-ESM instead of prescribing the topography and ice thickness. The planned set up for the ice-sheet model consists of a horizontal regular resolution of 10 km and the output fields are then remapped to 10' × 10'. Therefore, we decide to work in this first approach with the same kind of input data. Still, to obtain a better description of the bathymetric details in regions which might be critical for the ocean circulation and water masses changes, the TFD values were modified in few straits. For this purpose, we use the TFD from SRTM30_PLUS (Becker at al., 2009) for the Strait of Gibraltar, Bab-el-Mandeb, Denmark Strait, Faroe-Shetland Channel, Northwest Passage and Nares Strait. The obtained values were used to modify the TFD of the corresponding regions of the remapped RTopo-2 topography. The resulting field is our reference topography for present-day conditions.

For the generation of the GR30 bathymetry during the last deglaciation, we use the ICE-6G_C reconstructions (Argus et al., 2014; Peltier et al., 2015). They contain information on topography, orography and masks derived from a global model of glacial isostatic adjustment constrained by data. In particular, the variable called "topography" consists of values of ocean bathymetry on ocean points and the land/ice-sheet surface on land points. The variable called "topography difference from present" is the anomaly of topography respect to present. Variables called "land area fraction" and "ice area fraction" represent the land-sea mask and ice-sea mask, respectively. Finally, the grounded ice-mask can be derived by multiplying the land area fraction and the ice area fraction. The horizontal resolution is $1° \times 1°$ and the temporal resolution is 1 ka for the period spanning from 26 to 21 ka BP and 0.5 ka from 21 to 0 ka BP. Fields were interpolated to a $10' \times 10'$ regular grid.

In order to preserve the small-scale structures from the HR topography for a particular time slice, we use the anomalies from ICE-6G_C relative to present and add them to our present HR topography. This is applied in ice-free areas only. As the surface of an ice sheet is rather smoothed, we use here directly the bilinearly interpolated ICE-6G_C topography data where ice is grounded. The two data sets are merged using the interpolated grounded ice mask as weights yielding our HR-topography for a particular time slice.

### 3.1 Automatic generation of the GR30 bathymetry and land-sea mask

In this section, we describe the approach adopted to generate in an automatic way a bathymetry file to run MPIOM-GR30. Starting from a $10' \times 10'$ gridded topography (HR) as the input file, our script executes the following steps:

**(a)** Generation of the HR land-sea mask. First, a raw version of the land-sea mask (*rawLSM*) is generated using the values of the input topography, assigning 1 to the ocean or wet grids (negative values of topography) and 0 to the land or dry grids (positive or zero values of topography). The resulting *rawLSM* is modified to prevent small inland lakes and isolated wet grid points. The strategy is to keep only the wet points that are directly connected to one of the following basins: World Oceans, Mediterranean Sea, Red Sea, Black Sea and Caspian Sea. The wet points that are not connected to those basins are dried by assigning to them land-sea mask equal to 0. The result of this procedure is an HR land-sea mask in which only five basins are allowed to be wet and the smaller lakes are closed by assigning land to them.

**(b)** Generation of the GR30 land-sea mask. Reducing the resolution, in this case, can produce an unrealistic representation of the coastline due to the loss of details. Our strategy is to remap the HR land-sea mask to the GR30 MPIOM grid and then to modify it with focus on some specific features that are important for simulating the ocean circulation. We apply a first-order conservative remapping using the Climate Data Operator (CDO, 2015) to obtain values between 0 and 1 that we call "fraction ocean". The fraction ocean is, therefore, the fraction of the grid point that is wet. Then, we use that value to generate the GR30 land-sea mask taking into account the following aspects:

- A grid point is considered dry (land-sea mask equal to 0) if its value of fraction ocean is lower than 0.5.

• A grid point is considered wet (land-sea mask equal to 1) if its value of fraction ocean is larger or equal to 0.5 and if it is directly connected to one of the following basins: World Oceans, Mediterranean Sea, Red Sea, Black Sea and Caspian Sea. We apply here a similar approach as for the HR land-sea mask. Starting from one point in each basin, the wet area is expanded if the adjacent grid points have fraction ocean larger or equal to 0.5. The algorithm is then repeated until there is no point left that meets the former conditions.

• There might still exist grid points with fraction ocean larger or equal to 0.5 which are not considered wet by the previous step because they are not directly connected to any of the five basins. Thus, they represent isolated wet areas in the coarse grid. Because isolated lakes were prevented in the HR land-sea mask, we assume that they are artificially enclosed by the remapping and therefore they are forced to be connected to the open ocean. The fraction ocean is used to decide about the path of the connection, and the land grids with the largest fraction ocean are flooded (land-sea

mask to 1).

• Specific regions are considered in detail for further checking and the GR30 land-sea mask is, therefore, modified if necessary. First, we check if North and South America are connected by land or artificially separated by the remapping. Then, we check some straits or channels (Strait of Gibraltar, Bab-el-Mandeb, Bosphorus, Denmark Strait, Faroe-Shetland Channel, Northwest Passage, Nares Strait and the Strait of Sicily), islands (Indonesia and Japan) and

peninsulas (Florida, Thailand-Malaysia, Kamchatka, Italy and the Scandinavian Peninsula). The strategy here is to automatically control if the straits/channels are open or closed and if the islands/peninsulas are isolated from or connected to the mainland in the HR land-sea mask. To automatically perform this task, the algorithm finds the path of connection between two points apart. This is done in a restricted domain around the region of interest. For example, when checking the opening or closure of a strait, the points to be connected are wet points located in each side of the

strait. If the algorithm finds that the path of connection between both points is always within the ocean, that means that the strait is open. Instead, if the path of connection is blocked by land, that means that the strait is closed. The location of each pair of points was manually and carefully decided for each region and is fixed in the code. It was tested that those points do not change from wet to dry or vice versa during the last deglaciation. The approach is applied to each specific region mentioned before and both resolutions, HR and GR30. When necessary, the GR30 land-sea mask is

regionally modified to be consistent with the HR data. The information of the fraction ocean is used to decide about the path of the opening or closure. Being the fraction ocean a float number it is highly unlikely to obtain multiple solutions. In that case, the algorithm would choose the first solution found.

(c) Generation of the GR30 bathymetry. As we are only interested in ocean depth and want to exclude potential effects of mountains on land, we multiply the HR topography by the HR land-sea mask. This preserves the depth of ocean points

and sets land values to 0. Then, this HR ocean bathymetry is remapped to the GR30 MPIOM grid by applying a first-order conservative remapping. The resulting field is multiplied by the GR30 land-sea mask previously generated. Finally, the TFDs in some regions are modified according to the values of the HR bathymetry. The regions that are

checked are the same as in the previous step. This way, the artificial smoothing created by the remapping is corrected in order to guarantee an adequate TFD. As it was mentioned before, we assume that it is more important to get a good

representation of the TFD than of the area of the flow.

The resulting global GR30 bathymetry for present day is shown in Fig. 2b. Even though the resolution is coarse, the general features of both, the land-sea mask and depth compare well with the HR data (Fig. 2a). Two examples where the GR30 bathymetry is modified according to the values obtained for the HR one are plotted in Fig. 3. After remapping the land-sea mask, the Nares Strait resulted closed. This is an artificial effect because it appears open in the HR data (Fig. 3a). Therefore,

the GR30 bathymetry was modified in order to open the Nares Strait (Fig. 3b) and the TFD there was set to 167 m according to the HR value. In the case of the Denmark Strait, the TFD in the GR30 resulted in 549 m after the remapping. It was set to 600 m (Fig. 3d) in correspondence of the HR value (Fig. 3c) in order to obtain an improved representation of regional features.

## 3.2 Time-dependent GR30 bathymetry and land-sea mask

One important aspect to consider when the bathymetry is being changed within a simulation is to avoid sequences of rapid flooding and drying events of the shelves. We solve this issue by applying some resistance to change and thus, each new bathymetry field has a degree of dependency on the previous one. On the other hand, we limit the changes in both, ocean depth and land-sea mask in order to avoid abrupt transitions that can cause a model crash. As a result, changes in the ocean

configuration are slow enough to allow the model to run without numerical instability. Here we made a compromise between the speed of changes and the reproduction of realistic features. Therefore, the approach described in Sect. 3.1 requires an adaptation to be applied subsequently in time. For each time slice in which the ocean configuration is being changed, the approach for the generation of the GR30 bathymetry is similar to the one previously described. The differences are the following:

• Two files (instead of one) are needed as input data for executing the script. They are the HR field of the current time slice and the GR30 field of the previous time slice.

• The fraction ocean derived from the remapping (*rawFraction*) is then modified to get the final fraction ocean (*Fraction*) by taking into consideration the previous GR30 land-sea mask (*preMask*) through an inertia coefficient (*Icoeff*) as follows:

$$Fraction = \begin{cases} Min(rawFraction + Icoeff\,;1) \, if \ preMask = 1 \\ Max(0\,;rawFraction - Icoeff) \, if \ preMask = 0 \end{cases} \tag{1}$$

where *Icoeff* is a dimensionless coefficient and can have values between 0 and 1. Based on sensitivity analysis, we decide to use an *Icoeff* equal to 0.1. In this way, the resulting fraction ocean keeps, to some degree, memory of the previous land-sea mask.

- Changes in the land-sea mask are limited. New wet (dry) points are open (closed) only if they are directly connected to land (ocean) in the previous topography. New potential islands start being dried with one grid point.

- Changes in depth are restricted to the value $DH_{max}$ defined as:

$$DH_{max} = \begin{cases} dzw(1) \, ; \, if \, it \, is \, a \, new \, wet \, point \\ dzw(1) - 3 \, m \, ; \, elsewhere \end{cases} \tag{2}$$

where *dzw(1)* is the thickness of the first layer in MPIOM (15 m in our set up). The reason for this choice is that when accounting for the conservation of mass and tracers (Sect. 2.4.), large changes in depth can cause a negative thickness of the first layer of the model. This limitation does not affect the deep ocean where changes in bathymetry are slow but it can slow down the deepening or shallowing process of shelves.

To test the algorithms described in this section, the GR30 bathymetry from 21 ka BP to present day (forward in time) was generated. The HR dataset for the deglaciation previously described was interpolated in time to allow for the creation of a new GR30 bathymetry field every 10 years. The limitations in the changes of land-sea mask and depth are illustrated in Fig. 4 which shows time slices corresponding to the period when the Hudson Bay is being connected to the ocean. The black stars highlight the grid point where the process is initiated. The Hudson Bay is gradually opened and deepened by a slow process of flooding.

### 3.3 Matching changes in ocean volume and freshwater fluxes into the ocean

The growth or decay of ice sheets and the resulting net freshwater flux into the ocean is the only responsible mechanism to change the volume of the ocean in MPIOM, as incompressibility is assumed. Otherwise, effects like thermal expansion could be important as well. When running the model with a fixed bathymetry, the net freshwater fluxes into the ocean affect the mean sea surface height (SSH) and consequently the thickness of the uppermost ocean layer. When a new ocean bathymetry is derived in a formally independent process, the mass of water is distributed to the new configuration. Then, both estimates of the ocean volume should be consistent, and therefore, the mean SSH and mean thickness of the surface layer should be preserved within the simulation for all restart points. However, de facto, this is not always the case mainly for two reasons. On the one hand, the HR reconstructions might show inconsistencies if they do not exactly account for water conservation. On the other hand, reducing the resolution from HR to GR30 can cause disagreement in the ocean volume due to the loss of details in the bathymetry field. The aim of this step is to remove these two possible sources of inconsistencies. The procedure

is to match the last GR30 ocean volume with the ocean volume of the new GR30 configuration, by performing the following steps:

(a) The total volume of the ocean is computed before the generation of the new bathymetry ($V_{old}$). Both, the ocean depth and the modelled SSH are considered for the calculation.

(b) The new ocean bathymetry and land-sea mask are generated as described in Sect. 3.2. Then, the total ocean volume and
area of the new configuration are computed ($V_{new}$; $A_{new}$).

(c) The new depth in each grid point is modified by adding the constant value $C$ defined as:

$$C = \frac{\left(V_{old} - V_{new}\right)}{A_{new}} \qquad\qquad (3)$$

(d) The final changes in depth are again limited according to Eq. (2) to ensure model stability.

In this way, the resulting ocean GR30 bathymetry accounts for changes in the ocean volume due only to the freshwater
fluxes into the ocean. There might exist slight discrepancies produced by the last step. However, by removing possible artificial changes in ocean volume, the procedure ensures that the mean SSH is reasonably well preserved, independently of the freshwater fluxes and the prescribed HR dataset.

### 3.4 Adaptation of the restart file in order to conserve mass and tracers

The last modelled state of the ocean with its ocean configuration (restart file) will be used as the initial state for the later setup. Hence, the 2D and 3D fields should be adapted to the new bathymetry and land-sea mask. When carrying out this task, our aim is to account for the conservation of mass and tracers not only at global but also at regional scale. Therefore, the variables that are adapted in this step are SSH, sea ice, snow on sea ice (for conserving mass) and tracers (for conserving them). From here on, when referring to tracers, we mean temperature, salinity and any passive tracer that MPIOM
prognostically resolves (age tracer, radioactive tracer, CFC, etc.). The other model variables (like for example velocities) are not being modified. During the restart process, MPIOM multiplies the velocities with the land-sea mask, thus non-zero velocities are not a problem. However, on the coarse horizontal resolution applied in these very long climate model simulations, the velocities in the ocean are determined essentially by geostrophy and friction and after one month of simulation, the velocity field has adapted to the hydrographic fields. Our approach consists of the following steps:

(a) Vertical redistribution of water and tracers. In this first step, we keep the land-sea mask fixed and we only deal with changes in depth. 2D fields of SSH and 3D fields of tracers are vertically adjusted to the new depth. The strategy here is to conserve the volume and amount of tracers within the water column in each grid point. Considering an individual wet point, the SSH is modified according to the change in depth in order to preserve the ocean volume locally. For

example, consider a wet grid point in which the depth is 120.44 meters and the SSH from the restart file is -0.71 meters. The height of the water column results in 120.44 – 0.71 meters and the vertical levels for this configuration are shown in Fig. 5a. After changing the bathymetry, the depth at the same grid point is 122.16 meters. Because the grid area is unchanged, the SSH is lowered to -2.43 meters to conserve the volume of the water column and the vertical levels are adjusted as shown in Fig. 5b. As pointed out before, in MPIOM, the thickness of the uppermost or first layer depends on SSH, whereas the thickness of the deepest or last wet cell is adjusted to the bathymetry. The vertical distribution of all tracers is consistently moved along the vertical, taking into account the new layer thicknesses, in order to preserve the total amount of them within the water column. This involves the transfer of water and its properties between vertical adjacent boxes. The algorithm is formally identical to an advection using a first order upwind scheme with the depth changes being the product of time step and vertical velocity. The behaviour of the algorithms is displayed in Fig. 5 which shows an example of vertical profiles of temperature (Fig. 5c) and salinity (Fig. 5d). This way, the vertical profiles displayed in blue (Fig. 5c and d) are the ones from the original restart. The orange lines (Fig 5c and d) represent the original profiles shifted downward according to the change in depth. The resulting profiles after redistributing vertically the tracers to the new layer's thicknesses are displayed in green (Fig. 5d and d). Values of tracers are constant within each vertical layer of the model (stepped profile). As a result of deepening the bathymetry, the thickness of the bottom (surface) layer increase (decrease), whereas the middle layers remain unchanged (Figs. 5a and b). Therefore, to conserve tracers along the water column, vertical profiles are modified.

**(b)** Horizontal smoothing. The previous step is applied to each wet grid point independently, considering only changes in depth. Therefore, the resulting SSH field might present large gradients between adjacent grid points, which could lead to numerical instability when inserted into MPIOM. To fix this, the SSH field is smoothed by taking into consideration the conservation of mass and tracers. That is, when necessary, values of SSH are modified by moving a volume of water with its tracer properties between adjacent ocean grid points. The maximum permitted horizontal SSH gradient between neighbouring points is set to 0.2 meters, which seems to ensure numerical stability in the ocean model.

**(c)** Horizontal re-location of water, tracers, sea ice and snow on sea ice when the land-sea mask changes. In step (a) we describe the procedure for dealing with changes in depth only. In this step, the new wet (dry) points resulting from changes in the land-sea mask are filled (emptied). We avoid performing any kind of interpolation in this stage because it would not account for conservation of mass and tracers. Instead, in order to conserve properties, the necessary amount of water and tracers to fill new wet points is taken from other boxes. The simplest approach would be to take water from all ocean boxes. However, this would involve the artificial long-distance transfer of water mass properties. Therefore, we decide to use only adjacent ocean boxes. That is, small volumes of water with its properties coming from adjacent points is placed into the new wet point until completely filling it. Similarly, the amount of water and tracers from a point which is dried is re-located among the neighbouring wet grid points. This operation is repeated for sea ice and snow on sea ice. There needs to be a compromise between involving only a few neighbouring grid points and the

risk of obtaining large horizontal gradients of SSH. Sensitivity tests were performed to achieve the optimal balance for both, filling and emptying procedures.

**(d)** Horizontal smoothing. Again, we apply step (b) to obtain a sufficiently smooth SSH field to ensure numerical stability when running the model.

An example of the above-described approach is illustrated in Fig. 6, which shows regional SSH fields during an arbitrary time slice of the last deglaciation, when the north of Europe is still partially covered by ice. The SSH field from the original restart file is plotted in Fig. 6a. Then, the ocean bottom was deepened along the shelf due to a decay of the ice sheets. Accordingly, the SSH values after the vertical redistribution (Fig. 6b) are lower than the original ones (Fig. 6a). The land-sea mask also changed, there are a new wet and dry grid points (coloured in yellow and pink, respectively, in Fig. 6c). When drying a point, the amount of water is horizontally re-located among its neighbours resulting in a raised SSH in the surrounding area (red in Fig. 6c). Conversely, a lowered (blue in Fig. 6c) SSH field results when a grid point is flooded. A smoothing of the SSH field constitutes the last step to obtain the final field (Fig. 6d).

This approach guarantees the conservation of mass and tracers in the open ocean. The algorithms are not being applied in lakes (Caspian Sea and Black Sea when it is not connected to the open ocean) and therefore they constitute an exception for the conservation of water properties. MPIOM does not account for freshwater fluxes in lakes in order to avoid eventual over-flooding or drying them out. However, our algorithms could still be improved to take this aspect into consideration.

**4 Transient simulation**

This section has the aim of testing the above-described tool in a long-term run with MPI-ESM. The purpose is not to analyse the climate response to a changing bathymetry and land-sea mask, this will be discussed in a consecutive paper. The aim of this experiment is evaluating the performance of the tool in terms of model stability and conservation of mass and tracers. This is a necessary step towards a fully coupled simulation with interactive ice sheets.

We performed a simulation with MPI-ESM from 21 ka BP to 7 ka BP. The ICE6-G_C reconstructions were used to derive the HR topography as detailed in the beginning of Sect. 3. They were linearly interpolated in time to obtain changes in the bathymetry and land-sea mask every 10 years. In this way, the tool was applied 1400 times within the run and was tested under a wide range of conditions. The interpolated ICE6-G_C reconstructions were also used to compute the time-dependent freshwater fluxes into the ocean. First, the 10-year interval time derivative of the gridded ice thickness is calculated. Only the ice-sheet thicknesses at grounded points are considered. The time rate of change of this quantity is then divided by the density ratio between ice and freshwater to obtain the extra freshwater flux into the ocean:

$$F_{freshwater}(x,y,t) = \frac{-1}{R}\frac{\partial\,Ice(x,y,t)}{\partial t} \qquad (4)$$

where *Ice* is the ice thickness of the grounded-ice sheets and *R* the density ratio between ice and freshwater. The resulting value is considered constant for a period of 10 years, although it is introduced to the model every time step. The freshwater is transported into the ocean through the hydrological discharge (HD) model which considers the changes in river routing (Riddick et al., 2018). Hence, while the ice sheets melt, the ocean receives a positive net freshwater flux and the bottom topography and land-sea mask adapt to it. As a consequence, both the ocean volume and the ocean surface area increase in time. The relative changes of ocean volume and area along the simulation are plotted in Fig. 7. The ocean volume in the beginning and at the end of the run is equal to $1.2858 \times 10^{18}$ m$^3$ and $1.3313 \times 10^{18}$ m$^3$, respectively. These values represent a relative increase of approximately 3.58 % (Fig. 7a). The ocean surface area changed from $3.3692 \times 10^{14}$ m$^2$ to $3.6089 \times 10^{14}$ m$^2$ in the period of the simulation. This accounts for a relative change of 7.11 % (Fig. 7b) produced by changes in the land-sea mask. The rate of change increases from 14.5 ka BP onward, in response to the massive ice-sheet decay. There is a slight relative decrease in the ocean surface area by the end of the simulation, around 7200 yrs BP (Fig. 7b). This is because few grid points in the north of Canada were dried in that period due to uplift as consequence of glacial isostatic adjustment. Therefore, even though flooding events are dominant during the deglaciation, this particular case constitutes a test for drying points as well.

To illustrate the evolution of a prognostic variable computed within the ocean component of the MPI-ESM, Fig. 8 shows the sea surface temperature (SST, ºC) at different time slices. Changes in the land-sea mask can also be observed. For instance, the LGM (Fig. 8a) is characterized by the large extent of ice sheets considered as land by the model. The Black Sea is isolated and therefore, it is solved as a lake. Around 13 ka BP (Fig. 8b), the Scandinavian and the Siberian ice sheets are almost melted and the Antarctic ice sheet begins to retreat. The Laurentide ice sheet starts to melt and the Black Sea gets connected to the Mediterranean Sea around 10 ka BP (Fig. 8c). Conditions close to present-day are reached by the end of the simulation (Fig. 8d) when the Hudson Bay is open. The model is able to deal with a changing ocean configuration and computes the SST fields while the bathymetry and land-sea mask change.

The changes of ocean volume should match the freshwater fluxes into the ocean in order to account for water conservation. On the one hand, the differences in ocean volume derived from two consecutive restart files (10 years difference) were computed. We do not account for lakes in this calculation. The resulting time series is plotted in Fig. 9a (black line). On the other hand, for the same period of time, the freshwater fluxes into the ocean were integrated. This is, the monthly fields of freshwater input were multiplied by the grid area and by the number of seconds in that month. The resulting values were horizontally and temporally (each 10 years) integrated to obtain the total freshwater fluxes into the ocean in m$^3$ (Fig. 9a, red line). Both curves are almost identical indicating that changes in ocean volume are indeed only caused by the freshwater input. The difference between both time series was divided by the ocean area in order to obtain the errors in the mean sea level (Fig. 9b). They are of the order of $1 \times 10^{-3}$ cm and within the computational accuracy. Therefore, the changes in ocean volume match the freshwater input indicating that water is being conserved. Note that MPIOM uses the incompressibility assumption and therefore, the contribution of the thermal expansion on SSH is not being considered here. The year when the

Black Sea is connected to the Mediterranean Sea, around 10.3 ka BP, is an exception for the conservation. This is because in our approach we do not account for the conservation of water and tracers inside lakes. Therefore, when computing changes in the ocean volume, the water from the Black Sea is being introduced in the computation producing a large peak (Fig. 9a) that does not match with the freshwater input of that year.

Another aspect to check in the simulation is the conservation of water properties. Even if the ocean volume changes due to the freshwater input (Fig. 7), the global inventory of tracers should be constant in the absence of sources or sinks. The yearly mean of the global salt content was calculated for the period of the simulation using the model output written in 32 bits. Lakes are not being considered in the calculation. Figure 10 shows the relative change between two consecutive years. Relative errors are less than $1\times10^{-7}$ and can be considered within the computational error as it is the same accuracy for the model outputs. Because we use an identical approach for all the tracers (including salinity), we are confident that the global content of tracers is conserved in the long-term simulation with MPI-ESM. Again, the year when the Black Sea is connected to the Mediterranean Sea is an exception for the conservation of tracers.

As described in Sect. 3.3, the ocean bottom depth is adjusted in order to match changes in the ocean volume and the freshwater fluxes into the ocean. During this process, the mean SSH should remain unchanged. To evaluate the performance of the algorithms in dealing with this, the mean SSH was computed each time the restart files were modified. The calculation is done, therefore, every 10 years and the results are plotted in Fig. 11a. Deviations from a constant value are lower than 4 cm indicating that our approach is effective in maintaining the mean SSH unchanged. Figure 11b shows the mean SSH computed from the 10-years mean fields. The model runs for 10 years with the same ocean bathymetry and land-sea mask. Therefore, during that period, the ocean depth is not being adjusted to the unbalanced freshwater input. This causes changes in the mean SSH. Still, differences are low and the maximum deviation from zero is 30 cm (around 14.5 ka BP, Fig. 11b) in response to a very large freshwater flux. Considering that changing the bathymetry according to the freshwater fluxes corrects these inconsistencies (Fig. 11a), the tool could be applied more often within the simulation in order to improve the results. Nevertheless, we consider that these deviations are small and that a time step of 10 years is an optimal compromise between computing time and model performance.

## 5 Remarks

In this paper, we presented the strategy which we followed to automatically change the ocean bathymetry and land-sea mask in MPIOM, the ocean component of the MPI-ESM. The procedure for both, the generation of the bathymetry file and the adaptation of the restart file were described in details. The simulation presented here had the aim of evaluating the performance of the tool in terms of model stability and conservation of water properties. The algorithms showed very good behaviour for a long-term simulation and our approach guarantees the conservation of mass and tracers.

The principal tool consists of shell scripts that are called with a maximum of three input files. All the calculations are performed with CDO commands and programs written in FORTRAN. The tool can easily be included at the end of the main run script without the necessity of interrupting the simulation. There are two shell scripts that need to be called after the restart file is written by the model. The first one generates the new bathymetry file for running MPIOM. Two input files are required to run this script. The first one corresponds to a NetCDF file containing the new HR bathymetry. The second input

is an ASCII file which corresponds to the previous GR30 bathymetry as it was read by MPIOM. The output of this shell script is an ASCII file containing the new GR30 bathymetry to be read by the model. As a result, this script replaces the old bathymetry file to run MPIOM with the new one. The second shell script adapts the restart file generated by the model to the new ocean configuration. This script needs three input files. The first and second ones correspond to the old and new bathymetry files as read by MPIOM, respectively. The last input is the restart file generated by the model in NetCDF format.

The output is the modified restart file in NetCDF format to replace the original one. The execution of this tool needs the restart file generated by the model as input. Therefore, it can be called only after a restart file is generated. Contrary, it is possible to resubmit the job without applying the tool, that is with fixed bathymetry, land-sea mask and, therefore, unmodified restart file. This allows for a shorter number of years between resubmissions than the ones required for changing the bathymetry. Consequently, the tool is easy to apply and it is fast, taking less than a minute to run on a workstation.

There are mainly three limitations in our technique. First, the fact that changes in depth and coastlines are limited can slow down the flooding and drying events of the shelves. However, it is important to note that changes in topography in response to the ice-sheet retreat and isostatic adjustments are solved neither by the ocean model MPIOM nor by our algorithms. Instead, the HR topography is prescribed to our tool or solved by the ice-sheet model. In this sense, the non-linear changes or abrupt events that occurred during the last deglaciation are not affected by our methodology. Still, if the timing of the

flooding and drying events of the shelves is considered to be critical, the algorithms could be applied more often within the simulation (every year, for example). However, in MPI-ESM, changing the topography implies also changes in the river routing and the land mask for the atmospheric model. Therefore, there should be a compromise between the frequency that topography is being changed and the computational time. From our results, we conclude that changing the bathymetry every 10 years during the last deglaciation in our coarse resolution model is an optimal compromise between both, model

performance and computing time. Another possibility would be to widen the stencil used for collecting water for new ocean points. This would allow a faster propagation of coastlines by more than one grid point per iteration. This might also turn out to be necessary when applying the tool to ocean configurations with higher horizontal resolution. Second, this tool was originally written for the curvilinear orthogonal grid (GR) with two poles. Although we presented in this paper the results for the coarse resolution GR30, the tool can be also applied for the low resolution (GR15) configuration of MPIOM. Still, for

the moment its usage is limited to GR grids. We are currently working on a new version to include the tripolar (TP) quasi-isotropic grid (Murray, 1996) among the applications. In general, the algorithms are easily adapted to any ocean model that meets the same requirements as MPIOM: Arakawa-C grid in the horizontal, z-grid in the vertical including partial bottom

cells, free-surface and mass flux boundary conditions. However, there are some parameters inside the scripts that depend on the grid. They are the location of each pair of points in order to perform the checking steps described in Sect. 3.1 for correcting the bathymetric details. Third, our approach cannot guarantee the conservation of mass and tracers inside lakes. This is our first version and algorithms will be improved to include the lakes in the calculations, in particular, when the restart file is modified in order to fully conserve mass and tracers.

Despite the limitations mentioned above, this is, to our knowledge, the first time that changes in ocean bottom topography and coastlines are interactively computed in an ocean model for simulating the last deglaciation. Therefore, the presented modules constitute a step forward towards a realistic long-term simulation covering periods with strong topographic changes. We are currently continuing our efforts in the direction of an interactive coupling between MPI-ESM and the ice-sheet model. Our goal is to combine single components into a fully coupled ice sheet-solid earth-climate model with interactive coastlines and topography forced only with solar insolation and greenhouse gas concentrations.

## 6 Code and data availability

A version of the code is available under the 3-Clause BSD License on Zenodo at https://doi.org/10.5281/zenodo.1249579 (Meccia and Mikolajewicz, 2018). The MPI-ESM is available under the Software License Agreement version 2, after acceptance of a licence (https://www.mpimet.mpg.de/en/science/models/license/). The ICE-6G_C reconstructions used in this paper are freely accessible through the website: http://www.atmosp.physics.utoronto.ca/~peltier/data.php.

*Author contributions.* UM had the original idea, performed the experiment and coordinated the work. VLM developed the algorithms, made the analysis and figures and prepared the manuscript with contribution from UM.

*Competing interests.* The authors declare that they have no conflict of interest.

*Acknowledgements.* This work was funded by the German Federal Ministry of Education and Research (BMBF) through PalMod project, grant No. 01LP1513C. Special thanks to Anne Mouchet, Helmuth Haak and two anonymous reviewers for their comments and suggestions that helped to improve the manuscript.

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

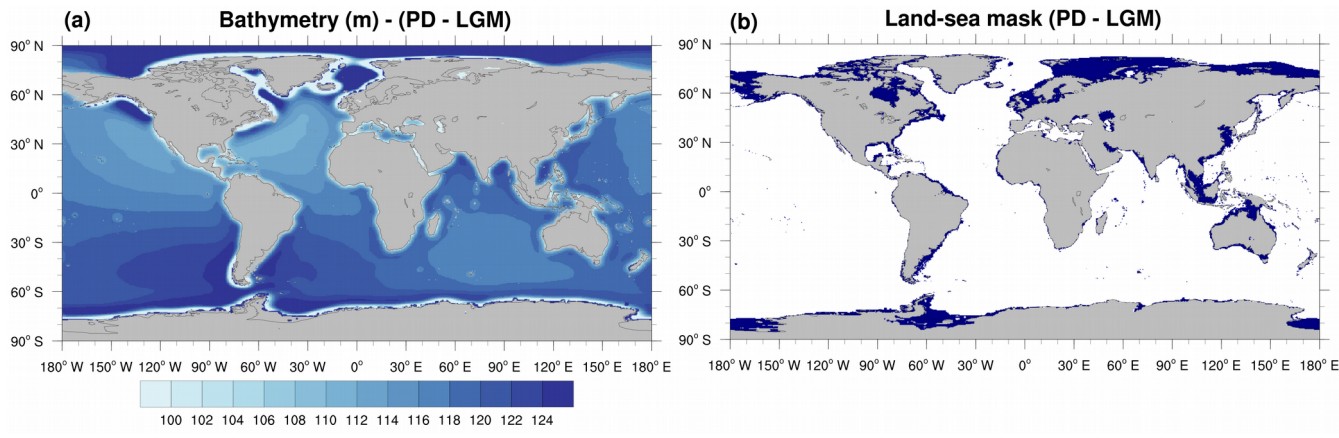

**Figure 1. Difference in (a) ocean bottom depth (m) and (b) land-sea mask between present-day conditions (PD) and 21 ka BP (LGM) estimated from the ICE-6G_C ice-sheet reconstructions. Blue in (b) represents ocean area during PD that were land during the LGM.**


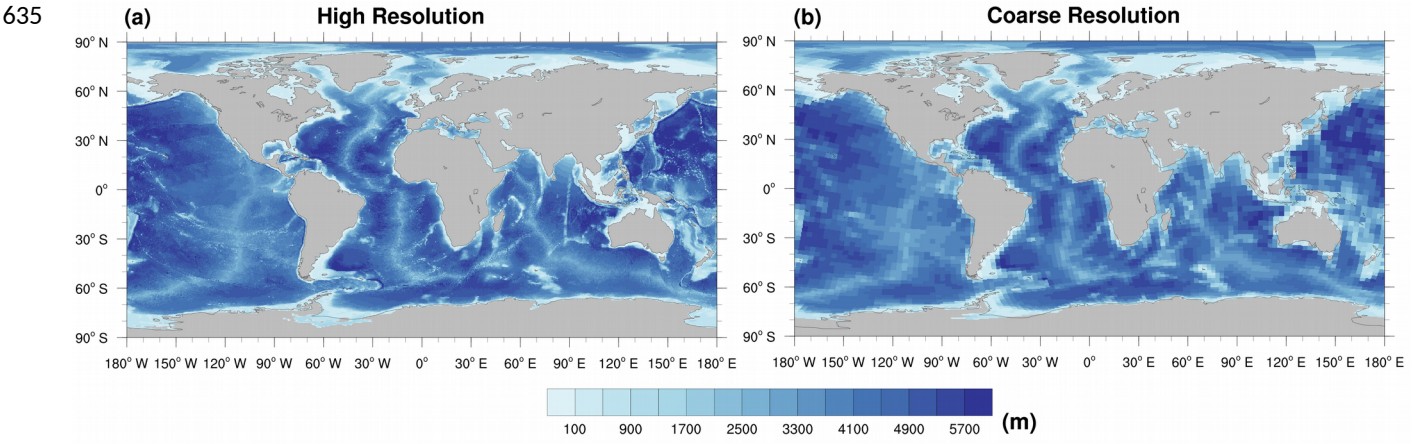

**Figure 2: Present-day global ocean bathymetry (m) and land-sea mask for (a) the high resolution (HR, 10' × 10') dataset; and (b) the generated coarse resolution (GR30) grid for running MPIOM.**

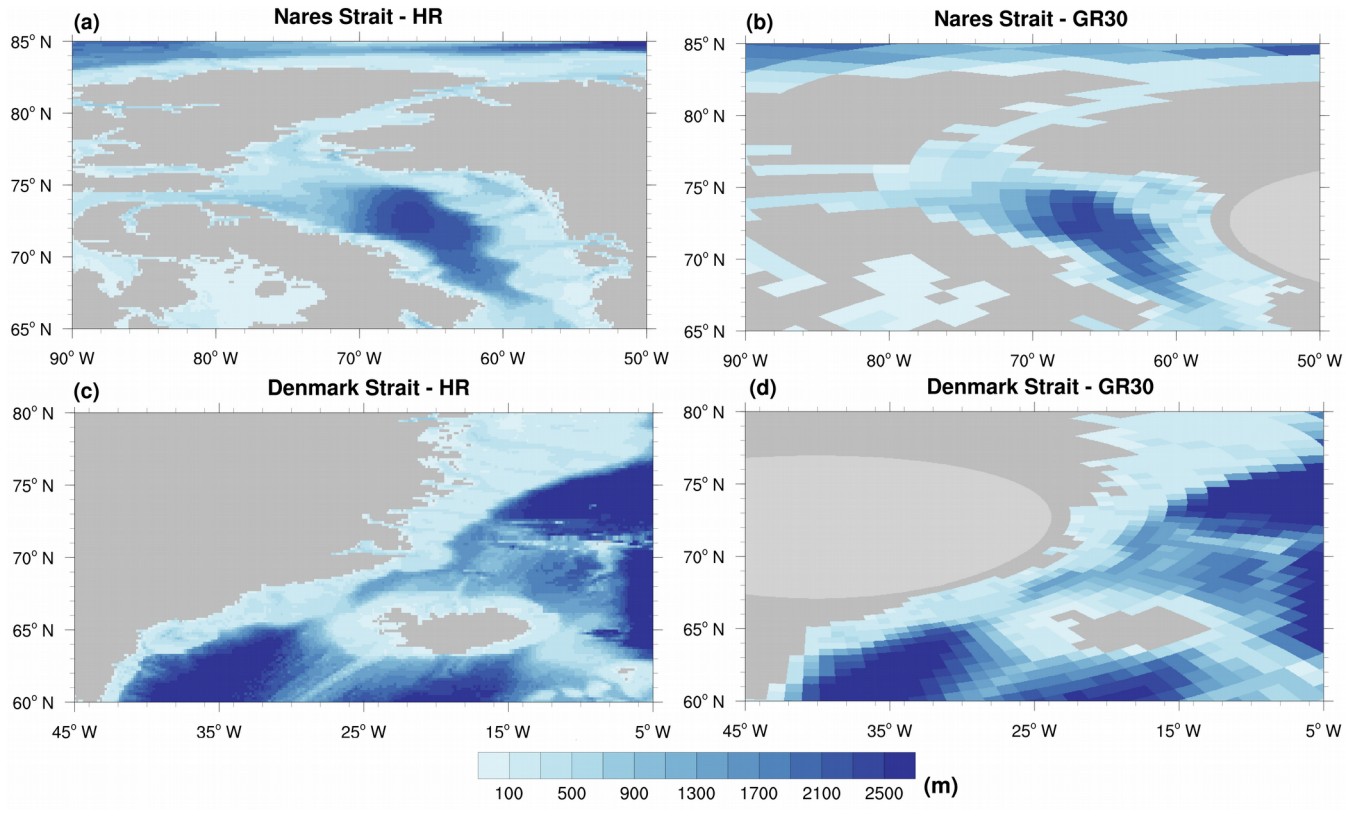

**Figure 3: Detailed present-day ocean bathymetry (m) and land-sea mask for the Nares Strait (a) HR; (b) GR30 and Denmark Strait (c) HR; (d) GR30.**

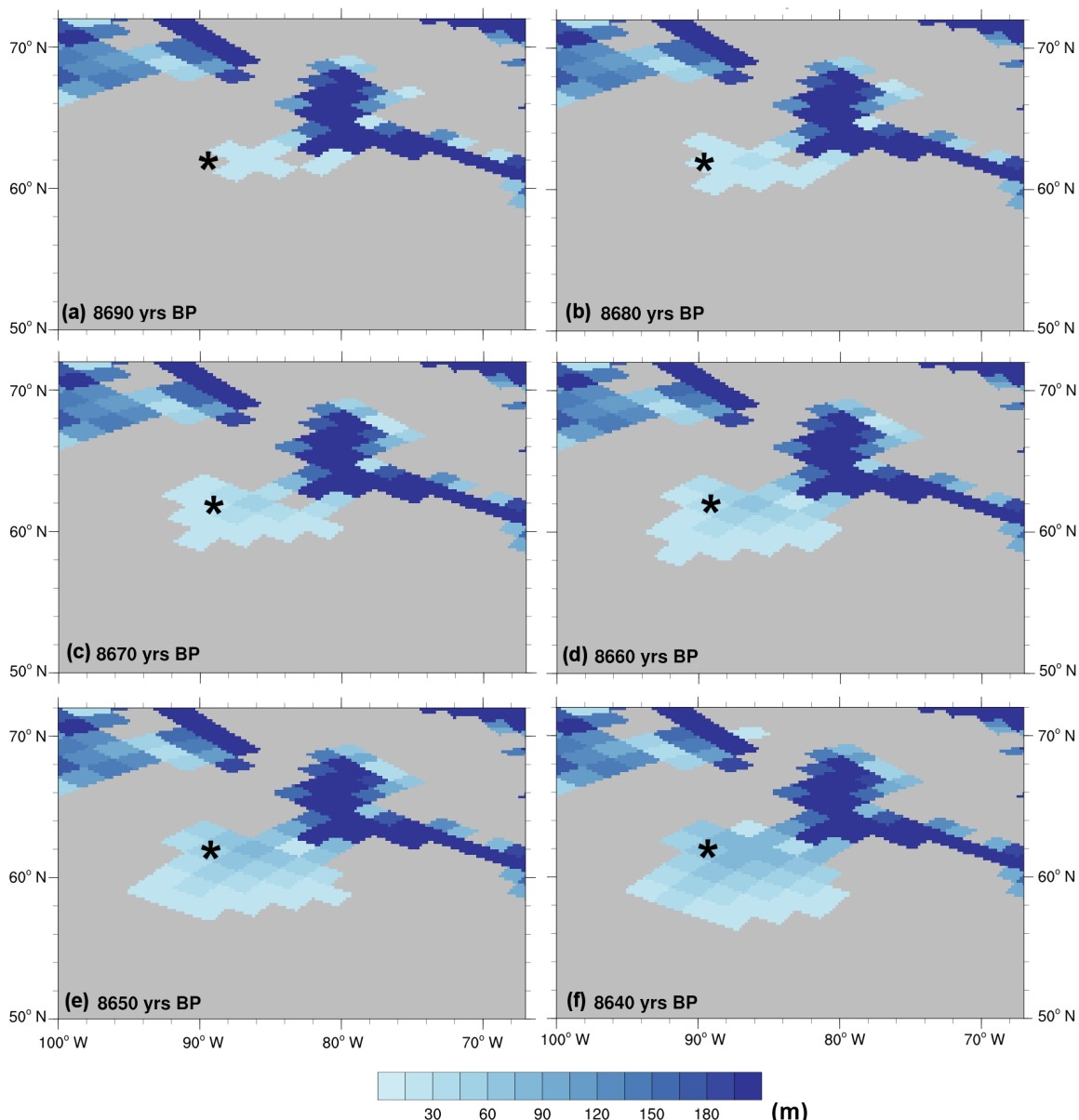

**Figure 4: Bathymetry fields around the Hudson Bay for different time slices (10 yrs time step) showing a gradual opening and deepening of the area. Black stars highlight a grid point as a reference.**



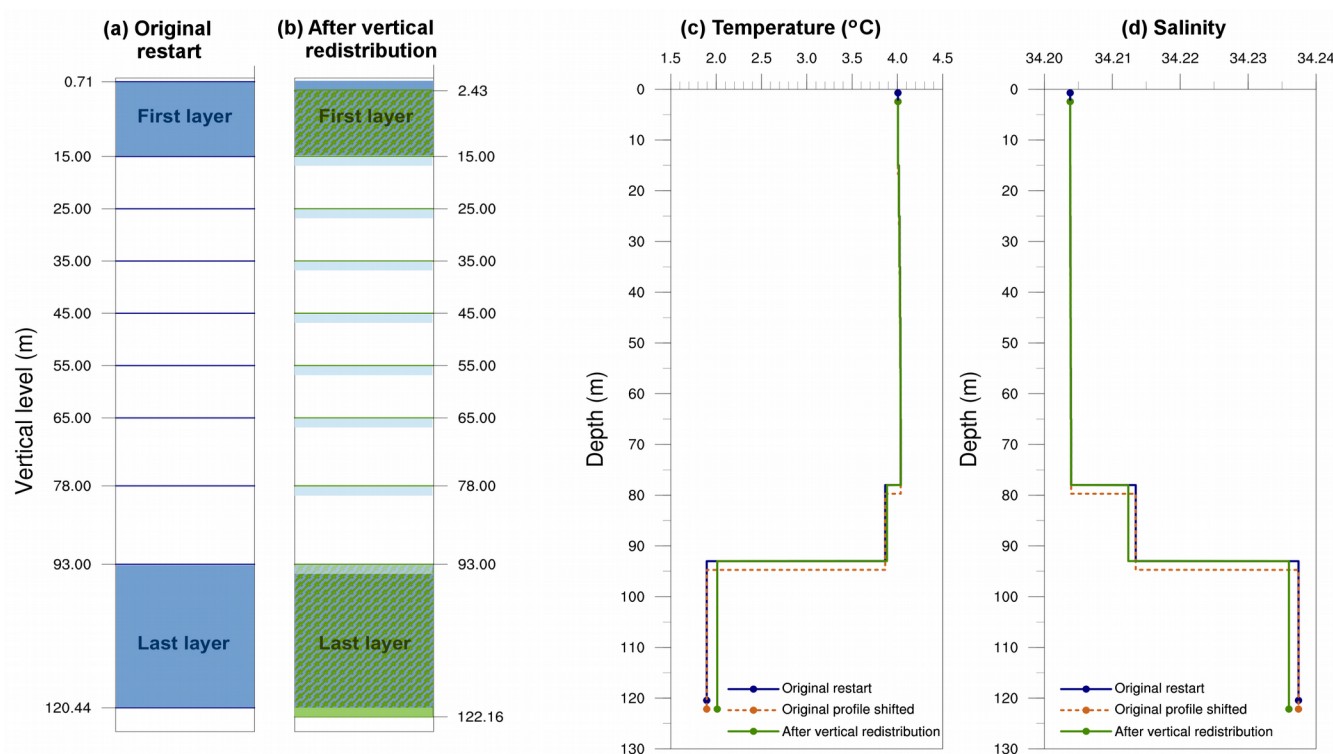

Figure 5: Example of vertical redistribution of water and tracers for a single wet grid point. Resulting vertical level configuration (a) before and (b) after changing the bathymetry. Blue and green areas represent the first and last vertical layer thicknesses from the original restart file and after the vertical redistribution, respectively. Hatched areas in (b) represent common thickness layer for both configurations. The light blue bars indicate where, after the vertical shift of the original profile, water from the layer above is added. Vertical profiles of (c) temperature and (d) salinity for the original restart fields (blue), the original profiles shifted downward according to the deepening in bathymetry (orange), and after applying the vertical redistribution in which the profiles are adapted to the model layers (green). Because values of tracers are constant within a model layer, the resulting profiles are stepped. Dots in (c) and (d) represent the upper limit of the first and the lower limit of the last vertical layer.

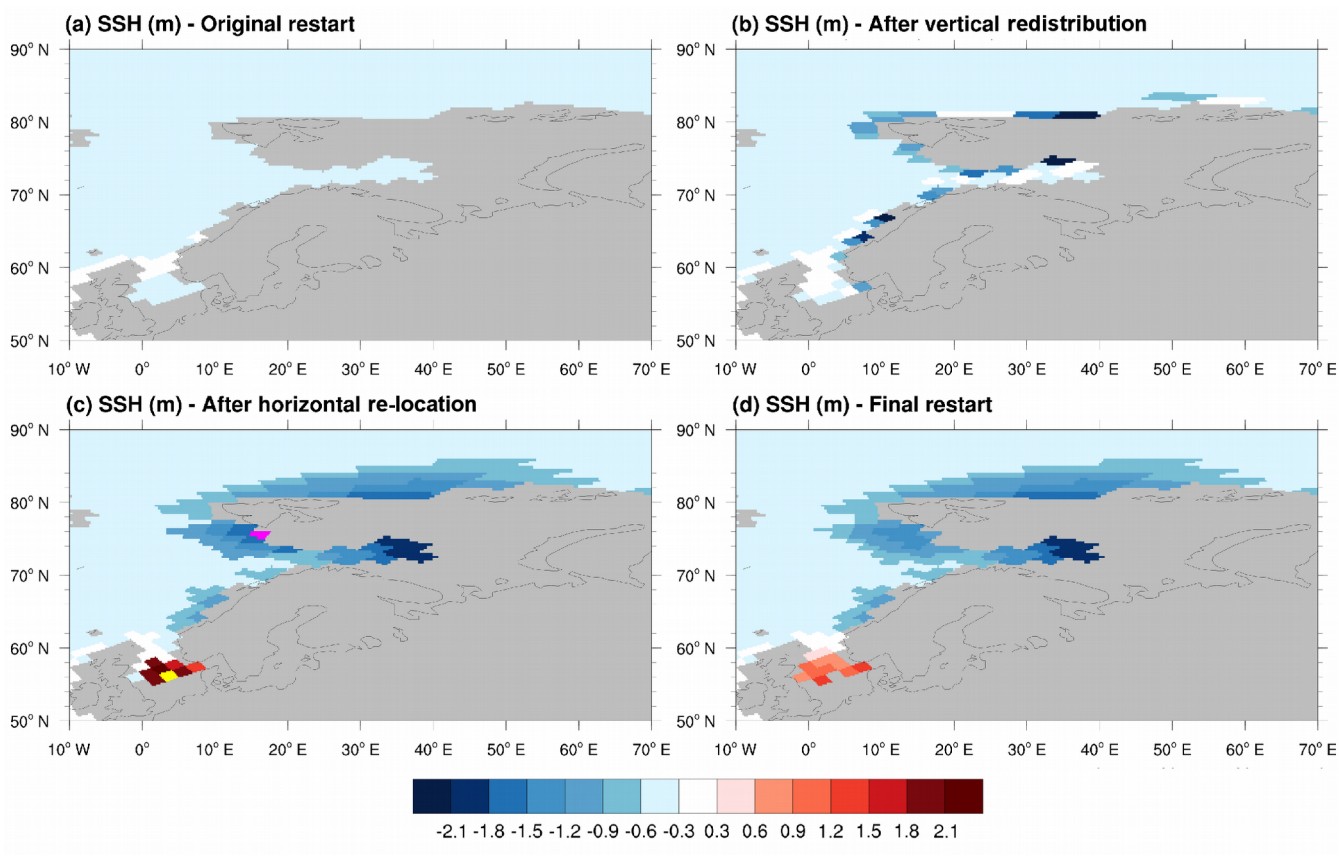

**Figure 6:** Example of the adaptation of the SSH field in order to conserve mass after changing the bathymetry. SSH field (m) (a) from the original restart file generated by MPIOM; (b) after the vertical redistribution; (c) after the horizontal re-location; and (d) after performing the horizontal smoothing. Grid points coloured in yellow and pink in (c) represent a new wet and dry points, respectively.

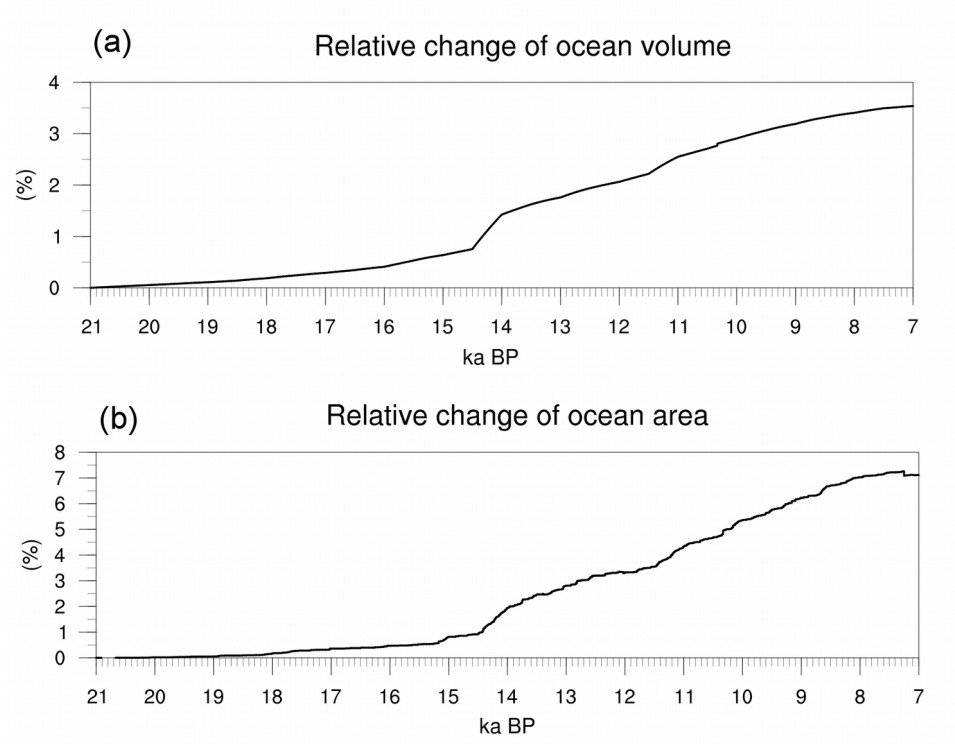

**Figure 7: Time series of relative change (%) with respect to the initial value for the computed yearly mean of (a) ocean volume; and (b) ocean surface area during the test run with MPI-ESM.**

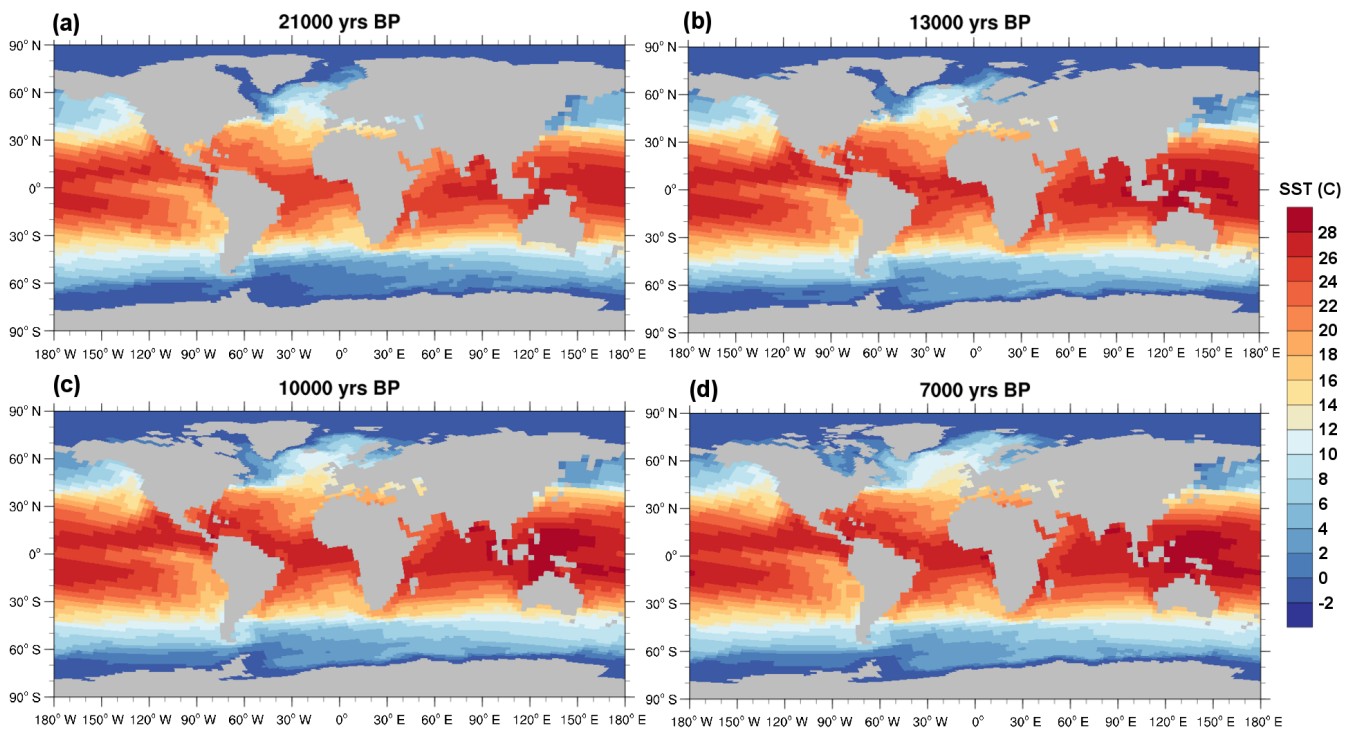

**Figure 8: Modelled SST (°C) for (a) 21; (b) 13; (c) 10; and (d) 7 ka BP. The model can resolve the new ocean points while the ice-sheets retreat.**

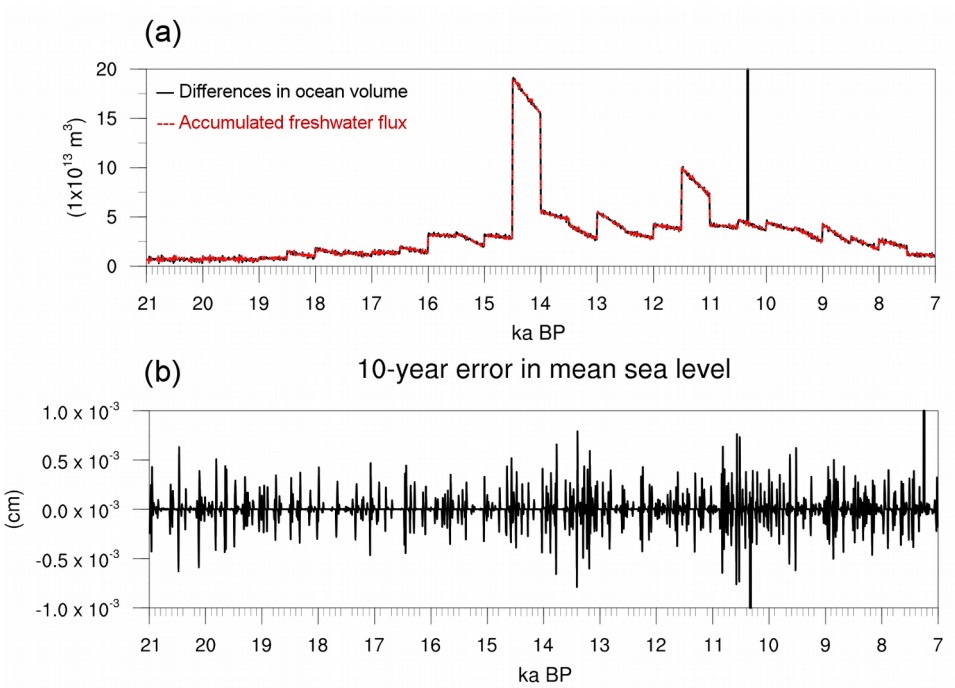

**Figure 9: Time series of (a) the 10-year differences in ocean volume (m³) derived from two consecutive restart files (black line) and 10-year accumulated freshwater fluxes into the ocean (m³, red line); and (b) difference (cm) between both divided by the ocean area during the test run with MPI-ESM.**

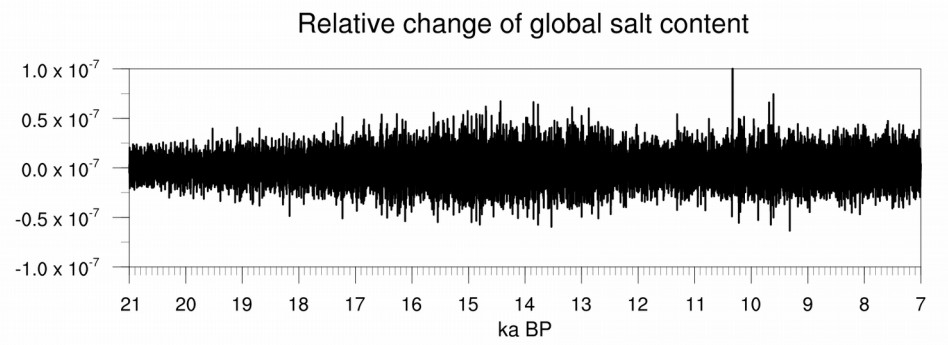

**Figure 10: Relative change of the yearly mean global salt content during the test run with MPI-ESM.**

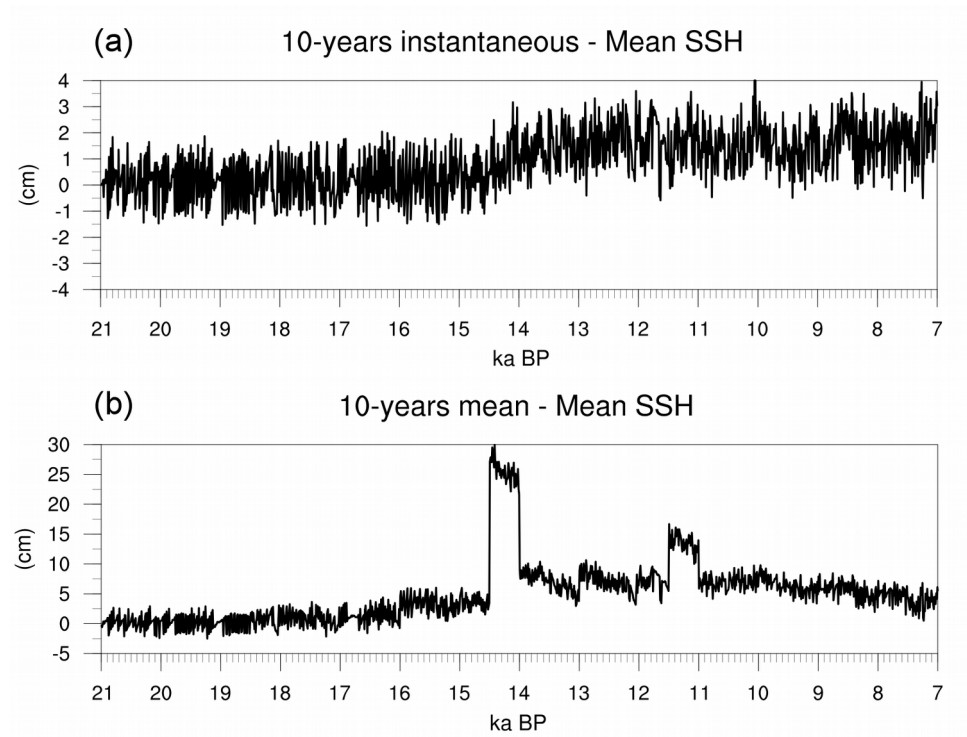

**Figure 11: Time series of (a) 10-years instantaneous; and (b) 10-years averaged mean SSH (cm) during the test simulation with MPI-ESM.**