# Peer review of "Interactive ocean bathymetry and coastlines for simulating the last deglaciation with the Max Planck Institute Earth System Model (MPI-ESM-v1.2)"

_Geoscientific Model Development, 2018_

## Referee Comment (RC1) · Anonymous Referee #1 · 3 Aug 2018

Review of GMD-2018-129
"Interactive ocean bathymetry and coastlines for simulating the last deglaciation with the Max Planck Institute Earth System Model (MPI-ESM-v1.2) by V. L. Meccia and U. Mikolajewicz for Geosci. Model Dev. Discuss.  Doi.org/10.5194/gmd-2018-129

Overall Comments:

This paper presents a complicated multi-step algorithm for automatically and successively modifying the MPI ocean model (MPIOM) bathymetry and land/ocean mask and restart input fields in a transient simulation under evolving boundary conditions such as for ice sheet growth and melt on long time scales. The set of time-stepped ICE6-G_C boundary conditions through the deglacial period are used here to demonstrate the utility and feasibility of the method. However, the ultimate goal is to be able to incorporate active solid earth and ice sheet models to drive ocean bathymetry, volume and coastline changes due to isostatic adjustments and added ice sheet meltwater fluxes that are important for simulating climate change over a glacial-interglacial cycle. This paper documents a new procedure for approaching an extremely challenging technical problem. Up to now, when ocean bathymetry or coastlines need to be changed over the course of a long transient simulation, it necessitates much human intervention and hands-on methods that may have been designed to be used once, and thus is usually done infrequently or not attempted at all.  The authors have demonstrated the success and feasibility of this new procedure that can automatically be applied at run-time and updated every 10 years for the long durations needed, though it is designed to be highly specific to MPIOM's particular model grid and architecture.  I recommend acceptance after some minor revisions that could help clarify the details of the procedure.

Specific comments:

1)  It should be mentioned in the procedural description that an important feature of the MPIOM is the employment of partial depth bottom cells, which makes their procedure possible. Models without partial bottom cells would be constrained to discrete values of bottom depth relative to the global mean sea surface (i.e., not including the sea surface height).

2)  Lines 107-112: This procedure omits any lakes that form other than those connected to the Caspian and Black seas.  The existence of large mid-continental post-glacial lakes formed following the melt and retreat at the southern boundary of the massive Laurentide ice sheet may be important for accurately reproducing the deglacial climate state. Drainage from Lake Agassiz, for example, and the routing of this significant source of meltwater to the ocean, is often hypothesized as causing changes in the meridional overturning circulation during the deglacial period. Excluding such lakes may be necessary in this first implementation of the tool, however, I suggest including a short explanation for why this step is required in this first implementation, the potential ramification, and plans for including them in the future.

3) As discussed again below, I found section 2.4, which describes the method for redistributing mass and tracers vertically and horizontally in the process of adjusting the restart files, difficult to follow. For example, what is meant by "vertical re-location" in line 259. A schematic diagram depicting the procedure following changes in depth would help to clarify this procedure.

4) There is no mention of what is done to adjust velocity components and other related fields that restart the flow fields following changes in land/ocean mask and bathymetry.

5) Section 3 describes how freshwater fluxes are added to the ocean from the melt of grounded ice sheets by the river discharge model, effectively increasing ocean volume. Section 2 describes the procedure for changing bathymetry and ocean volume using the ICE6-G_C data, implicitly changing volume due to melting ice sheets. Figure 8 shows the procedure works out as the volume change from these two processes match, but it reads like the ocean volume is being changed twice here. Is it because the bathymetry changes are made as a result of the meltwater added slowly over previous interval of time (10 years) since last bathymetric changes? Thus, new volume, added through bathymetry changes, lags or catches up to the volume change due to freshwater added through meltwater over the preceding interval? A schematic showing all of these complicated steps would help clarify.

Minor comments by line:
63: "In the frame of the project…" -- awkward phrase to start the sentence.

150: OK--omitting Arctic and Southern Oceans in this list because they are contiguous with the other major ocean basins?

171:"Specific regions are examined in detail and modified if necessary." Also, "…look at the HR land-sea mask…" Suggests human oversight here, but I suspect this is not the case. This step must use some rather specialized coding because many specific regions in the HR mask are checked against the new GR30 mask. What methods are used to make this more automatic? Are multiple solutions possible to obtain using the fraction ocean in the new GR30 to identify pathways that connect new regions?

Section 2.3 and section 3, lines 300-308: Globally adjusting ocean depth to keep global mean SSH constant, and volume changes through adding freshwater from melting ice sheets. Are the steps employed in Section 2.3, done every timestep after freshwater from melting grounded ice sheets is added, thus increasing global ocean volume through increases in SSH?

248: Do the final changes made to depth as described in step 2.3(d) require iterations back to (3)?

259: Section 2.4(a) What is meant by "vertical re-location"? I find this section difficult to understand the actual details of the method even after looking at Figure 5. A schematic

illustrating the method would be helpful, especially for locations that are already "wet" points that become deeper. How are tracers at mid-depth changed? Also does the process of vertical re-location" result in lateral gradients at depth in the ocean, even after horizontal smoothing? 263: "new layer's thickness"?

300: The "instantaneous time derivative of the gridded ice thickness" is computed for the meltwater fluxes added by the hydrological discharge model. Does this gridded ice sheet thickness come from the ICE6-G_C data interpolated in time to every 10 years? Does "instantaneously" mean the meltwater flux calculation is done every time step, or for every 10-year interval?

366: "…called with a maximum of three input files." The description of the tool software and scripts is short. A bit more information about how it is used in practice and integrated into the model run-time would be helpful. For example, how does it interface with the model during run-time? Which input files are needed? Is the tool launched in the main run script, at the start of a restart submission using files from the previous submission? Because restart files are generated, does this mean that the 10 year time interval between bathymetry changes fixes the maximum number of years between resubmission?

---

## Referee Comment (RC2) · Anonymous Referee #2 · 6 Aug 2018

Review of the paper entitled "Interactive ocean bathymetry and coastlines for simulating the last deglaciation with the Max Planck Institute Earth System Model (MPI-ESM-v1.2)" by Virna Loana Meccia and Uwe Mikolajewicz.

This paper has been long awaited  by the community working on the last deglaciation from LGM to present day. But, in fact, this methodology could also be interesting for simulations for future deglaciations of Greenland and West Antarctica in the next century.

Indeed in the framework of PMIP4 deglaciation project (Ivanovic 2016) in which models intend to provide transient simulations from LGM to PD, such tool is absolutely needed.

The authors aim to use the MPI ESM to produce deglaciation transient runs. They cope with a long lasting issue and resolve it: how to modify boundary conditions that account for sea level rise during the deglaciation and modify the topography (bathymetry and coastal lines) all along this process using algorithms that avoid manual and more or less subjective corrections. They describe the algorithms they used for adaptation of the ocean model MPIO at low resolution used in the PMIP4 exercise with boundary conditions evolving every 10 year.

The paper is well written and its structure is clear. The detailed description of strategy target and problems is convincing.

My major comments are the following:

1 the paper is perfectly suited for GMD. Nevertheless the authors never tackle the effect of their boundary condition changes on deglaciation. Therefore I suggest that they address this question at least concerning two important points

- Discussing the added value of this study compared to previous simulations where the bathymetry was not changed to better emphasize what may be the interest of this study beyond the technical challenges.
- The authors should also emphasize the potential limitations of this method in terms of simulating abrupt events during deglaciation due to many linear processes they used, both in time and space. I perfectly understand smoothing procedures the authors described to avoid crash of the model, but during deglaciation many non linear changes occurred for instance MPW and more generally acceleration of melting rates described for instance in C. Waelbroeck et al., Quaternary Science Reviews 21, 295-305, 2002, for the  last 30k. Therefore the authors should discuss in more details what is the compromise between avoiding crash and capturing real non linear events.

2. The authors should also clarify the part of the paper that may be directly useful for the PMIP4 deglaciation community and those that have been developed specifically for MPI ESM.
Whereas this paper is worth to be published in GMD, I have also minor comments that it would be important the authors answer to before publication.

Minor comments:

Abstract:

A1: What do the authors mean by conservation of mass and tracers at regional scale. It is a bit misleading in the abstract. I think the authors have in mind to keep regional conservation when changing spatial resolution. They should clarify this issue.

A2 The authors, first tackle a very general problem: the bathymetry adaptation when simulating the last deglaciation. How far the algorithm developed here, beyond grid specificity can be easily adapted to other models. A sentence in the abstract should clarify this point.

Introduction

I1 The first sentence is very general and partially untrue because of some aspects of the unprecedented speed of ongoing climate change. The authors should remove or modify this sentence.

I2 The authors should mention that major uncertainties remain on reconstruction of Antarctica at LGM. Indeed, NH ice sheet reconstructions are better constrained, whereas Antarctica ice sheet reconstruction has often been an adjustable parameter. Therefore, the authors should mention Antarctica reconstruction uncertainties at LGM both from data and models (G. Philippon, Earth and Planetary Science Letters 248 (2006) 750.)

I3 The authors should also mention that there have been already many successful publications on glacial-interglacial simulations cycles from EMIC (A. Ganopolski et al., Nature 529, pages 200–203 (2016)) and from GCM (A. Abe-Ouchi et al., Nature 500, (2013)190. Moreover, the authors should better emphasize what in this context would be the added value of accounting for sea level rise.

I4 Superimposed to the vertical resolution of MPIO, an important issue to be discussed is the choice of the initial horizontal resolution.

Methodology:

M1 It is not clear for me that accounting for only two big lakes (Caspian and Black Sea), the authors can capture abrupt climate changes occurring during deglaciation, as for instance the 8.2 ka event. Moreover, the evolution of Caspian and Black Sea associated to Eurasian ice-sheet melting and large modification of the catchment is not easy to be reconstructed and depicted. The authors should clarify more explicitly what is the limit of their method. Specifically, they should explain how they cope with river run-off and changes in catchment areas during deglaciation for these two epicontinental seas. These issues have been shown to have drastic consequences on atmosphere and ocean circulation (see for example R. Alkama et al., GRL 33 (21) 2006, R. Alkama et al., 2008, Climate Dynamics. 30 and M. Wary et al, J. Quaternary Sci. 32, 908–922, 2017).

M2 At the end of paragraph 2.3, in the spatial smoothing procedure for SSH, there are also changes in water mass reorganization that lead to spatial variations of the sea level rise during melting as shown for instance in Mitrovica (Nature 2001,…). Is this effect accounted for? If not, the authors should clarify the possible impact of this process.

Results:

R1: Whereas this paper is submitted for publication in GMD and devoted to technical and model development aspects, it is difficult to consider the validity of the process only analyzing the stability of the response without any information on the potential climate effect. Indeed, accounting for bathymetry with time steps of 10 years should allow the authors to capture the complex pattern of the deglaciation periods. Nevertheless, due to linear smoothing in time and space, it is unclear to me whether they really may capture abrupt events. This limitation should be discussed in more details.

R2: Superimposed to ice sheet melting, a major component of the SLR is the ocean thermal expansion during deglaciation. Therefore it should produce a difference between SLR and cumulative fresh water input. In fig. 8, I suggest to plot, superimposed to the black and red curves, the component relative to the changes of the ocean volume associated with the thermal expansion during deglaciation.

R3: is the model accounting for a possible ice shelf at the beginning of the deglaciation in the northern hemisphere?

Remarks:

RM1 As the impact on climate due to change in bathymetry is not described in this paper, we can still have in mind many questions concerning the limits of this tool, when applied to non linear processes as those occurring during deglaciation. Indeed, the deglaciation is far to be a linear process. Major abrupt events (MWP and HE) occurred that are associated with large increase of fresh water inputs. It would be interesting that the authors discuss these potential limitations.

Final comment:

This study is interesting and novel. Moreover, it corresponds to an awaited development to better simulate the last transient deglaciation. Therefore when the authors will have answered the questions raised above, the manuscript will be worth to be published.

---

## Author Comment (AC1)

Reply to Referee #1, by V.L. Meccia and U. Mikolajewicz

**Review of GMD-2018-129**

"Interactive ocean bathymetry and coastlines for simulating the last deglaciation with the Max Planck Institute Earth System Model (MPI-ESM-v1.2) by V. L. Meccia and U. Mikolajewicz for Geosci. Model Dev. Discuss. Doi.org/10.5194/gmd-2018-129

**Overall Comments:**

This paper presents a complicated multi-step algorithm for automatically and successively modifying the MPI ocean model (MPIOM) bathymetry and land/ocean mask and restart input fields in a transient simulation under evolving boundary conditions such as for ice sheet growth and melt on long time scales. The set of time-stepped ICE6-G C boundary conditions through the deglacial period are used here to demonstrate the utility and feasibility of the method. However, the ultimate goal is to be able to incorporate active solid earth and ice sheet models to drive ocean bathymetry, volume and coastline changes due to isostatic adjustments and added ice sheet meltwater fluxes that are important for simulating climate change over a glacial-interglacial cycle. This paper documents a new procedure for approaching an extremely challenging technical problem. Up to now, when ocean bathymetry or coastlines need to be changed over the course of a long transient simulation, it necessitates much human intervention and hands-on methods that may have been designed to be used once, and thus is usually done infrequently or not attempted at all. The authors have demonstrated the success and feasibility of this new procedure that can automatically be applied at run-time and updated every 10 years for the long durations needed, though it is designed to be highly specific to MPIOM's particular model grid and architecture. I recommend acceptance after some minor revisions that could help clarify the details of the procedure.

We thank Referee #1 for his/her useful comments. We give a detailed response to each issue in what follows.

**Specific comments:**

1) It should be mentioned in the procedural description that an important feature of the MPIOM is the employment of partial depth bottom cells, which makes their procedure possible. Models without partial bottom cells would be constrained to discrete values of bottom depth relative to the global mean sea surface (i.e., not including the sea surface height).

We include in the manuscript a section in which the model requirements are described:

**"2 Ocean model requirements**

The algorithms presented in this paper are tailored for the coarse resolution setup of MPIOM but should be easily transferable to other model resolutions or other ocean models having similar assumptions and approximations. MPIOM is a free-surface ocean general circulation model with the hydrostatic and Boussinesq approximations and incompressibility is assumed. It solves the primitive equations on an Arakawa-C grid in the horizontal and a z-grid in the vertical (Maier-Reimer 1997). For freshwater, a mass-flux boundary condition is implemented. A detailed description of the model equations and its physical parametrizations is given in Marsland et al. (2003) while its performance as the ocean component of the MPI-ESM is evaluated by Jungclaus et al. (2013). MPIOM includes an embedded dynamic/thermodynamic sea-ice model (Notz et al., 2013) with a viscous-plastic rheology following Hibler (1979). Sea-ice is swimming in the water. Ice shelves are not included. In this paper,

we use the MPIOM coarse resolution configuration with a curvilinear orthogonal grid (GR30) and two poles (Haak et al., 2003), over Greenland and Antarctica. We decide to use the coarse configuration to reduce the computational time, but the algorithms presented in this paper can easily be adapted to higher resolution grids. In the vertical, the model has 40 unevenly spaced levels, ranging from 15 meters near the surface to several hundred meters in the deep ocean. Vertical discretization includes partial vertical grid cells. Therefore, at each horizontal grid point, the deepest wet cell has a thickness that is adjusted to resolve the discretized bathymetry. On the other hand, the surface layer thickness is also adjusted to account for the sea surface elevation and the sea ice/snow where appropriate."

2) Lines 107-112: This procedure omits any lakes that form other than those connected to the Caspian and Black seas. The existence of large mid-continental post-glacial lakes formed following the melt and retreat at the southern boundary of the massive Laurentide ice sheet may be important for accurately reproducing the deglacial climate state. Drainage from Lake Agassiz, for example, and the routing of this significant source of meltwater to the ocean, is often hypothesized as causing changes in the meridional overturning circulation during the deglacial period. Excluding such lakes may be necessary in this first implementation of the tool, however, I suggest including a short explanation for why this step is required in this first implementation, the potential ramification, and plans for including them in the future.

Our algorithms are applied within the ocean model and therefore they work on the ocean domain. You are right that mid-continental post-glacial lakes are important for reproducing the deglacial climate state. But, actually, this is a problem that should be treated in the land-model instead of the ocean one. As a matter of fact, including such lakes when considering changes in the routing of the meltwater to the ocean is an ongoing work (as a follow-up of Riddick et al., 2018). For the ocean model, the freshwater fluxes into the ocean is a forcing and the algorithms presented in this paper do not treat the problem of how that forcing is derived.

Because we are solving only the ocean domain, we are interested only in lakes that are connected to the ocean, that is the Black Sea. The Caspian Sea is, indeed, an exemption because it is not connected to the oceans. However, the Caspian Sea is much larger than the other minor lakes. We decided to include it to solve the SST there that might impact on the climate of Central Asia. Therefore, solving the SST of the Caspian Sea might be important for coupled climate models.

We clarify this issue at the beginning of section 2 (now 3) Methodology:

"Finally, we check for the presence of lakes in the GR30 bathymetry; the Caspian Sea and the Black Sea (under LGM condition, for example) are the only cases that are permitted. Because we are dealing with an ocean model, we are interested in lakes that are connected to the ocean, that is the Black Sea. However, we include the Caspian Sea in our calculations because of its potential impact on the climate of Central Asia. Solving the SST of the Caspian Sea, which is much larger than other minor lakes, might be important for coupled climate simulations. All other lakes need to be removed from the ocean domain either by connecting them to the open ocean or by considering them as land. The atmospheric model component allows accounting for lakes on land (only the thermal component). In the framework of our model system, the adequate place to calculate water storage in lakes is the hydrological discharge model."

**3) As discussed again below, I found section 2.4, which describes the method for**

redistributing mass and tracers vertically and horizontally in the process of adjusting the restart files, difficult to follow. For example, what is meant by "vertical re-location" in line 259. A schematic diagram depicting the procedure following changes in depth would help to clarify this procedure.

Thanks for this comment. We realize that we were not clear enough and we reformulate part of section 2.4 (now 3.4) Adaptation of the restart file in order to conserve mass and tracers:

"Our approach consists of the following steps:

(a) Vertical redistribution of water and tracers. In this first step, we keep the land-sea mask fixed and we only deal with changes in depth. 2D fields of SSH and 3D fields of tracers are vertically adjusted to the new depth. The strategy here is to conserve the volume and amount of tracers within the water column in each grid point. Considering an individual wet point, the SSH is modified according to changes in depth in order to preserve the ocean volume locally. For example, consider a wet grid point in which the depth is 120.44 meters and the SSH from the restart file is -0.71 meters. The height of the water column results in 120.44 - 0.71 meters and the vertical levels for this configuration are shown in Fig. 5a. After changing the bathymetry, the depth at the same grid point is 122.16 meters. Because the grid area is unchanged, the SSH is lowered to -2.43 meters to conserve the volume of the water column and the vertical levels are adjusted as shown in Fig. 5b. As pointed out before, in MPIOM, the thickness of the uppermost or first layer depends on SSH, whereas the thickness of the deepest or last wet cell is adjusted to the bathymetry. The vertical distribution of tracers is consistently moved along the vertical, taking into account the new layers thickness, in order to preserve the total amount of them within the water column. The behaviour of the algorithms is displayed in Fig. 5 which shows an example of vertical profiles of temperature (Fig. 5c) and salinity (Fig. 5d). This way, the vertical profiles displayed in blue (Fig. 5c and d) are the ones from the original restart. The orange lines (Fig 5c and d) represent the original profiles shifted downward according to the change in depth. The resulting profiles after redistributing vertically the tracers to the new layer's thicknesses are displayed in green (Fig. 5d and d). Values of tracers are constant within each vertical layer of the model (stepped profile). As a result of deepening the bathymetry, the thickness of the bottom (surface) layer increase (decrease), whereas the middle layers remain unchanged (Figs. 5a and b). Therefore, to conserve tracers along the water column, vertical profiles are modified.

---

## Author Response (AR1)

[revised manuscript text omitted]

**Overall Comments:**
**This paper presents a complicated multi-step algorithm for automatically and successively modifying the MPI ocean model (MPIOM) bathymetry and land/ocean mask and restart input fields in a transient simulation under evolving boundary conditions such as for ice sheet growth and melt on long time scales. The set of time-stepped ICE6-G_C boundary conditions through the deglacial period are used here to demonstrate the utility and feasibility of the method. However, the ultimate goal is to be able to incorporate active solid earth and ice sheet models to drive ocean bathymetry, volume and coastline changes due to isostatic adjustments and added ice sheet meltwater fluxes that are important for simulating climate change over a glacial-interglacial cycle. This paper documents a new procedure for approaching an extremely challenging technical problem. Up to now, when ocean bathymetry or coastlines need to be changed over the course of a long transient simulation, it necessitates much human intervention and hands-on methods that may have been designed to be used once, and thus is usually done infrequently or not attempted at all. The authors have demonstrated the success and feasibility of this new procedure that can automatically be applied at run-time and updated every 10 years for the long durations needed, though it is designed to be highly specific to MPIOM's particular model grid and architecture. I recommend acceptance after some minor revisions that could help clarify the details of the procedure.**

We thank Referee #1 for his/her useful comments. We give a detailed response to each issue in what follows.

**Specific comments:**
**1) It should be mentioned in the procedural description that an important feature of the MPIOM is the employment of partial depth bottom cells, which makes their procedure possible. Models without partial bottom cells would be constrained to discrete values of bottom depth relative to the global mean sea surface (i.e., not including the sea surface height).**

We include in the manuscript a section in which the model requirements are described:

" 2 Ocean model requirements
The algorithms presented in this paper are tailored for the coarse resolution setup of MPIOM but should be easily transferable to other model resolutions or other ocean models having similar assumptions and approximations. MPIOM is a free-surface ocean general circulation model with the hydrostatic and Boussinesq approximations and incompressibility is assumed. It solves the primitive equations on an Arakawa-C grid in the horizontal and a z-grid in the vertical (Maier-Reimer 1997). For freshwater, a mass-flux boundary condition is implemented. A detailed description of the model equations and its physical parametrizations is given in Marsland et al. (2003) while its performance as the ocean component of the MPI-ESM is evaluated by Jungclaus et al. (2013). MPIOM includes an embedded dynamic/thermodynamic sea-ice model (Notz et al., 2013) with a viscous-plastic rheology following Hibler (1979). Sea-ice is swimming in the water. Ice shelves are not included. In this paper,

we use the MPIOM coarse resolution configuration with a curvilinear orthogonal grid (GR30) and two poles (Haak et al., 2003), over Greenland and Antarctica. We decide to use the coarse configuration to reduce the computational time, but the algorithms presented in this paper can easily be adapted to higher resolution grids. In the vertical, the model has 40 unevenly spaced levels, ranging from 15 meters near the surface to several hundred meters in the deep ocean. Vertical discretization includes partial vertical grid cells. Therefore, at each horizontal grid point, the deepest wet cell has a thickness that is adjusted to resolve the discretized bathymetry. On the other hand, the surface layer thickness is also adjusted to account for the sea surface elevation and the sea ice/snow where appropriate."

**2) Lines 107-112: This procedure omits any lakes that form other than those connected to the Caspian and Black seas. The existence of large mid-continental post-glacial lakes formed following the melt and retreat at the southern boundary of the massive Laurentide ice sheet may be important for accurately reproducing the deglacial climate state. Drainage from Lake Agassiz, for example, and the routing of this significant source of meltwater to the ocean, is often hypothesized as causing changes in the meridional overturning circulation during the deglacial period. Excluding such lakes may be necessary in this first implementation of the tool, however, I suggest including a short explanation for why this step is required in this first implementation, the potential ramification, and plans for including them in the future.**

Our algorithms are applied within the ocean model and therefore they work on the ocean domain. You are right that mid-continental post-glacial lakes are important for reproducing the deglacial climate state. But, actually, this is a problem that should be treated in the land-model instead of the ocean one. As a matter of fact, including such lakes when considering changes in the routing of the meltwater to the ocean is an ongoing work (as a follow-up of Riddick et al., 2018). For the ocean model, the freshwater fluxes into the ocean is a forcing and the algorithms presented in this paper do not treat the problem of how that forcing is derived.

Because we are solving only the ocean domain, we are interested only in lakes that are connected to the ocean, that is the Black Sea. The Caspian Sea is, indeed, an exemption because it is not connected to the oceans. However, the Caspian Sea is much larger than the other minor lakes. We decided to include it to solve the SST there that might impact on the climate of Central Asia. Therefore, solving the SST of the Caspian Sea might be important for coupled climate models.

We clarify this issue at the beginning of section *2 (now 3) Methodology*:

"Finally, we check for the presence of lakes in the GR30 bathymetry; the Caspian Sea and the Black Sea (under LGM condition, for example) are the only cases that are permitted. Because we are dealing with an ocean model, we are interested in lakes that are connected to the ocean, that is the Black Sea. However, we include the Caspian Sea in our calculations because of its potential impact on the climate of Central Asia. Solving the SST of the Caspian Sea, which is much larger than other minor lakes, might be important for coupled climate simulations. All other lakes need to be removed from the ocean domain either by connecting them to the open ocean or by considering them as land. The atmospheric model component allows accounting for lakes on land (only the thermal component). In the framework of our model system, the adequate place to calculate water storage in lakes is the hydrological discharge model."

**3) As discussed again below, I found section 2.4, which describes the method for**

**redistributing mass and tracers vertically and horizontally in the process of adjusting the restart files, difficult to follow. For example, what is meant by "vertical re-location" in line 259. A schematic diagram depicting the procedure following changes in depth would help to clarify this procedure.**

Thanks for this comment. We realize that we were not clear enough and we reformulate part of section *2.4 (now 3.4) Adaptation of the restart file in order to conserve mass and tracers*:

"Our approach consists of the following steps:

[revised manuscript text omitted]

Thus, a figure was added to the revised manuscript and figure numbering has changed accordingly.

**4) There is no mention of what is done to adjust velocity components and other related fields that restart the flow fields following changes in land/ocean mask and bathymetry.**

The aim of adapting the restart file when bathymetry and land-sea mask change is to account for the conservation of mass and tracers. Therefore, the modification of the fields is done for sea surface height, sea-ice, snow on sea ice (because they are key variables for the total ocean mass), temperature, salinity and passive tracers (because we want to conserve tracers). We do not perform any computation for the other variables (including velocity components) and, therefore, their values remain unmodified. Thus, when wetting a new grid point, the values of the restart file for velocity, for example, will be zero because it is the value for a dry/land point. During the restart procedure MPIOM anyway guarantees that velocity on land points is set to zero. Considering the horizontal resolution that is currently being applied in long simulations with climate models, the advection of momentum is of minor importance. Far from the equator, velocity can be approximated pretty well by frictional geostrophy, as done in the LSG model (Maier-Reimer et al., 1993). Even though velocity is formally a prognostic variable of the ocean model, it is de facto a diagnostic variable whereas the main prognostic ones are temperature and salinity and sea ice. This fact is exploited in typical set-up procedures, where the ocean is initialized with fields of temperature and salinity (from climatology or other model runs) and at rest. However, after one month the velocity field is adapted to the hydrographic fields.

We mention that in section *2.4 (now 3.4) Adaptation of the restart file in order to conserve mass and tracers*:

"The last modelled state of the ocean with its ocean configuration (restart file) will be used as the initial state for the later setup. Hence, the 2D and 3D fields should be adapted to the new bathymetry and land-sea mask. When carrying out this task, our aim is to account for the conservation of mass and tracers not only at global but also at regional scale. Therefore, the variables that are adapted in this step are SSH, sea-ice, snow on sea ice (for conserving mass) and tracers (for conserving them). From here on, when referring to tracers, we mean temperature, salinity and any passive tracer that MPIOM prognostically resolves (age tracer, radioactive tracer, CFC, etc.). The other model variables (like for example velocities) are not being modified. During the restart process, MPIOM multiplies the velocities with the land-sea mask, thus non-zero velocities are not a problem. However, on the coarse horizontal resolution applied in these very long climate model simulations, the velocities in the ocean are determined essentially by geostrophy and friction and after one month of simulation, the velocity field has adapted to the hydrographic fields. Our approach consists of the following steps:"

**5) Section 3 describes how freshwater fluxes are added to the ocean from the melt of**

**grounded ice sheets by the river discharge model, effectively increasing ocean volume. Section 2 describes the procedure for changing bathymetry and ocean volume using the ICE6-G_C data, implicitly changing volume due to melting ice sheets. Figure 8 shows the procedure works out as the volume change from these two processes match, but it reads like the ocean volume is being changed twice here. Is it because the bathymetry changes are made as a result of the meltwater added slowly over previous interval of time (10 years) since last bathymetric changes? Thus, new volume, added through bathymetry changes, lags or catches up to the volume change due to freshwater added through meltwater over the preceding interval? A schematic showing all of these complicated steps would help clarify.**

Ocean volume is not being changed twice. Section 3 describes the transient simulation we performed in order to test the algorithms. The ICE6-G_C reconstructions were used to derive the HR topography and to compute the time-dependent freshwater fluxes into the ocean as in eq. (4). This step is necessary here because the HR topography is prescribed in this experiment and will not be needed when coupling the climate model with the ice-sheet and solid earth models. The ice-sheet growth or decay and the resulting net freshwater flux into the ocean is the only responsible process for the changes in ocean volume and ocean surface area (fig. 6). Therefore, the changes in ocean volume should match the net freshwater fluxes into the ocean (fig. 8), except for the delay caused by the time needed within the hydrological discharge model to transport the water to the ocean.

When running the model for 10 years with a fixed bathymetry, the imbalance of net freshwater fluxes into the ocean affects the mean SSH, which is simply a consequence that during the 10 year simulation the ocean volume has changed and is not matching exactly the bathymetry any more. After 10 years, the bathymetry and land-sea mask change and the mass of water is distributed to the new configuration. This way, the mean SSH is being preserved within the simulation. This would work perfectly if a) the model ocean bathymetry had the same horizontal resolution than the topography used to compute the freshwater fluxes into the ocean; b) the data used for computing the freshwater fluxes and HR topography was consistent because it accounts for conservation of water. However, a) reducing the resolution from HR to GR30 results in a smoother bathymetry that might result in differences in ocean volume (between the one that would be for HR and the one that results for GR30); b) we aim at writing an algorithm independently of the data used as forcing and so we consider the potential inconsistencies in the reconstructions. Hence, the aim of step 2.3 is to correct these two possible sources of inconsistencies. The strategy is to match the last ocean volume state (GR30, which already accounts for the accumulated freshwater fluxes during the previous 10 years) with the ocean volume of the new configuration (GR30 that might contain artificial changes in ocean volume due to the loss of bathymetric details when reducing resolution and to potential inconsistencies in the HR topography). The resulting ocean GR30 bathymetry accounts for changes in the ocean volume only due to the imbalanced net freshwater fluxes.

We understand that this issue is not clear enough in the original manuscript and we modify section *2.3 (now 3.3) Matching changes in ocean volume and freshwater fluxes into the ocean*:

"The growth or decay of ice sheets and the resulting net freshwater flux into the ocean is the only responsible mechanism to change the volume of the ocean in MPIOM, as incompressibility is assumed. Otherwise, effects like thermal expansion could be important as well. When running the model with a fixed bathymetry, the net freshwater fluxes into the ocean affect the mean SSH and consequently the thickness of the uppermost ocean layer. When a new ocean bathymetry is derived in a formally independent process, the mass of water is distributed to the new configuration. Then, both estimates of

the ocean volume should be consistent, and therefore, the mean SSH and mean thickness of the surface layer should be preserved within the simulation for all restart points. However, de facto, this is not always the case mainly for two reasons. On the one hand, the HR reconstructions might show inconsistencies if they do not account for water conservation. On the other hand, reducing the resolution from HR to GR30 can cause disagreements in the ocean volume due to the loss of details in the bathymetry field. The aim of this step is to remove these two possible sources of inconsistencies. The procedure is to match the last GR30 ocean volume, which already accounts for the freshwater fluxes into the ocean, with the ocean volume of the new GR30 configuration, by performing the following steps:"

and later in the same section:

"In this way, the resulting ocean GR30 bathymetry accounts for changes in the ocean volume due only to the freshwater fluxes into the ocean. There might exist slight discrepancies produced by the last step. However, by removing possible artificial changes in ocean volume, the procedure ensures that the mean SSH is reasonably well preserved, independently of the freshwater fluxes and the prescribed HR dataset."

**Minor comments by line:**
**63: "In the frame of the project..." -- awkward phrase to start the sentence.**

Reformulated to:

"Our long-term goal in the context of the project "From the Last Interglacial to the Anthropocene: Modeling a Complete Glacial Cycle – (PalMod)", is to simulate the last termination with a coupled ice sheet-solid earth-climate model with interactive coastlines and topography forced only with solar insolation and greenhouse gases concentration."

**150: OK--omitting Arctic and Southern Oceans in this list because they are contiguous with the other major ocean basins?**

Yes. Atlantic-Pacific-Indian Oceans was changed to World Oceans:

"The strategy is to keep only the wet points that are directly connected to one of the following basins: World Oceans, Mediterranean Sea, Red Sea, Black Sea and Caspian Sea."

Accordingly, it also was changed in line 163 of the original manuscript.

**171:"Specific regions are examined in detail and modified if necessary." Also, "...look at the HR land-sea mask..." Suggests human oversight here, but I suspect this is not the case. This step must use some rather specialized coding because many specific regions in the HR mask are checked against the new GR30 mask. What methods are used to make this more automatic? Are multiple solutions possible to obtain using the fraction ocean in the new GR30 to identify pathways that connect new regions?**

Yes, this step is done automatically by the code. We explain it in the revised manuscript:

"Specific regions are considered in detail for further checking and the GR30 land-sea mask is, therefore, modified if necessary. First, we check if North and South America are connected by land or

artificially separated by the remapping. Then, we check some straits or channels (Strait of Gibraltar, Bab-el-Mandeb, Bosphorus, Denmark Strait, Faroe-Shetland Channel, Northwest Passage, Nares Strait and the Strait of Sicily), islands (Indonesia and Japan) and peninsulas (Florida, Thailand-Malaysia, Kamchatka, Italy and the Scandinavian Peninsula). The strategy here is to automatically control if the straits/channels are open or closed and if the islands/peninsulas are isolated from or connected to the mainland in the HR land-sea mask. To automatically perform this task, the algorithm finds the path of connection between two points apart. This is done in a restricted domain around the region of interest. For example, when checking the opening or closure of a strait, the points to be connected are wet points located in each side of the strait. If the algorithm finds that the path of connection between both points is always within the ocean, that means that the strait is open. Instead, if the path of connection is blocked by land, that means that the strait is closed. The location of each pair of points was manually and carefully decided for each region and is fixed in the code. It was tested that those points do not change from wet to dry or vice versa during the last deglaciation. The approach is applied to each specific region mentioned before and both resolutions, HR and GR30. When necessary, the GR30 land-sea mask is regionally modified to be consistent with the HR data. The information of the fraction ocean is used to decide about the path of the opening or closure. Being the fraction ocean a float number it is highly unlikely to obtain multiple solutions. In that case, the algorithm would choose the first solution found."

**Section 2.3 and section 3, lines 300-308: Globally adjusting ocean depth to keep global mean SSH constant, and volume changes through adding freshwater from melting ice sheets. Are the steps employed in Section 2.3, done every timestep after freshwater from melting grounded ice sheets is added, thus increasing global ocean volume through increases in SSH?**

In the transient simulation we performed, the freshwater fluxes into the ocean are being incorporated every time step, whereas the procedure described in section 2.3 is being applied only for constructing a new ocean bathymetry, in this case, every 10 years. We clarify this aspect in the revised manuscript. Please, see the answer to your point 5) of "*Specific comments*" for more details.

**248: Do the final changes made to depth as described in step 2.3(d) require iterations back to (3)?**

No. It is important for the model stability that the final depth satisfies eq. (2). For example, a new wet point must have a depth smaller than the thickness of the surface layer in the model. This is to avoid involving more than one layer when adapting the restart file. Going back to this criteria might destroy some corrections done in step 2.3 (now 3.3) as stated in the manuscript:

 "In this way, the resulting ocean GR30 bathymetry accounts for changes in the ocean volume due only to the freshwater fluxes into the ocean. There might exist slight discrepancies produced by the last step. However, by removing possible artificial changes in ocean volume, the procedure ensures that the mean SSH is reasonably well preserved, independently of the freshwater fluxes and the prescribed HR dataset."

Yet, we demonstrated that water is being conserved in a long-term transient simulation (fig. 8) indicating that, if the discrepancies still exist, they are not large enough to affect the mass conservation.

**259: Section 2.4(a) What is meant by "vertical re-location"? I find this section difficult to**

**understand the actual details of the method even after looking at Figure 5. A schematic illustrating the method would be helpful, especially for locations that are already "wet" point that become deeper. How are tracers at mid-depth changed? Also does the process of vertical re-location" result in lateral gradients at depth in the ocean, even after horizontal smoothing? 263: "new layer's thickness"?**

We better explained the methods and we added a new figure in the revised manuscript. Please, see the answer to your point 3) of "*Specific comments*" for more details.

**300: The "instantaneous time derivative of the gridded ice thickness" is computed for the meltwater fluxes added by the hydrological discharge model. Does this gridded ice sheet thickness come from the ICE6-G_C data interpolated in time to every 10 years? Does "instantaneously" mean the meltwater flux calculation is done every time step, or for every 10-year interval?**

Yes, the gridded ice sheet thickness comes from the interpolation in time to every 10 years. The derivative is done once for every 10 years interval. We clarify this in the revised manuscript:

"The interpolated ICE6-G_C reconstructions were also used to compute the time-dependent freshwater fluxes into the ocean. First, the 10-year interval time derivative of the gridded ice thickness is calculated. Only the ice-sheet thicknesses at grounded points are considered. The time rate of change of this quantity is then divided by the density ratio between ice and freshwater to obtain the extra freshwater flux into the ocean:
*eq. (4)*
where *Ice* is the ice thickness of the grounded-ice sheets and *R* the density ratio between ice and freshwater. The resulting value is considered constant for a period of 10 years, although it is introduced to the model every time step. The extra freshwater is transported into the ocean through a hydrological discharge (HD) model which considers the changes in river routing (Riddick et al., 2018)."

**366: "...called with a maximum of three input files." The description of the tool software and scripts is short. A bit more information about how it is used in practice and integrated into the model run-time would be helpful. For example, how does it interface with the model during run-time? Which input files are needed? Is the tool launched in the main run script, at the start of a restart submission using files from the previous submission? Because restart files are generated, does this mean that the 10 year time interval between bathymetry changes fixes the maximum number of years between resubmission?**

A more detailed description is included in the revised manuscript:

"The principal tool consists of shell scripts that are called with a maximum of three input files. All the calculations are performed with CDO commands and programs written in FORTRAN. The tool can easily be included at the end of the main run script without the necessity of interrupting the simulation. There are two shell scripts that need to be called after the restart file is written by the model. The first one generates the new bathymetry file for running MPIOM. Two input files are required to run this script. The first one corresponds to a NetCDF file containing the new HR bathymetry. The second input is an ASCII file which corresponds to the previous GR30 bathymetry as it was read by MPIOM. The output of this shell script is an ASCII file containing the new GR30 bathymetry to be read by the model. As a result, this script replaces the old bathymetry file to run MPIOM with the new one. The second shell script adapts the restart file generated by the model to the new ocean configuration. This

script needs three input files. The first and second ones correspond to the old and new bathymetry files as read by MPIOM, respectively. The last input is the restart file generated by the model in NetCDF format. The output is the modified restart file in NetCDF format to replace the original one. The execution of this tool needs the restart file generated by the model as input. Therefore, it can be called only after a restart file is generated. Contrary, it is possible to resubmit the job without applying the tool, that is with fixed bathymetry, land-sea mask and, therefore, unmodified restart file. This allows for a shorter number of years between resubmissions than the ones required for changing the bathymetry. Consequently, the tool is easy to apply and it is fast, taking less than a minute to run on a workstation."

Reply to Referee #2, by V.L. Meccia and U. Mikolajewicz

**Review of the paper entitled "Interactive ocean bathymetry and coastlines for simulating the last deglaciation with the Max Planck Institute Earth System Model (MPI-ESM-v1.2)" by Virna Loana Meccia and Uwe Mikolajewicz.**

**This paper has been long awaited by the community working on the last deglaciation from LGM to present day. But, in fact, this methodology could also be interesting for simulations for future deglaciations of Greenland and West Antarctica in the next century.**

**Indeed in the framework of PMIP4 deglaciation project (Ivanovic 2016) in which models intend to provide transient simulations from LGM to PD, such tool is absolutely needed.**

**The authors aim to use the MPI ESM to produce deglaciation transient runs. They cope with a long lasting issue and resolve it: how to modify boundary conditions that account for sea level rise during the deglaciation and modify the topography (bathymetry and coastal lines) all along this process using algorithms that avoid manual and more or less subjective corrections. They describe the algorithms they used for adaptation of the ocean model MPIO at low resolution used in the PMIP4 exercise with boundary conditions evolving every 10 year.**

**The paper is well written and its structure is clear. The detailed description of strategy target and problems is convincing.**

We thank Referee #2 for his/her useful comments. We give a detailed response to each issue in what follows.

**My major comments are the following:**

**1. the paper is perfectly suited for GMD. Nevertheless the authors never tackle the effect of their boundary condition changes on deglaciation. Therefore I suggest that they address this question at least concerning two important points**

We propose this paper to GMD as a "Development and technical paper" because it presents a novel methodology consisting of several steps tackling a challenging technical problem. We believe that a detailed description of the algorithms deserves a paper itself and therefore we are submitting a purely technical paper. Thus, we are not aiming at analyzing the climate response to a changing bathymetry and land-sea mask in this study. The effects of including our algorithms in a transient simulation of the last deglaciation in terms of the climate response will be faced in another paper and it is an ongoing work.

- **Discussing the added value of this study compared to previous simulations where the bathymetry was not changed to better emphasize what may be the interest of this study beyond the technical challenges.**

  Discussing the added value of applying the algorithms described in the manuscript in comparison to a simulation in which the bathymetry and land-sea mask are fixed is for sure a very interesting and necessary issue. However, as mentioned before, it is not the aim of our manuscript and it will be the topic of another paper. We believe that the effects of including a variable topography for simulating the last deglaciation deserve a detailed study itself. Indeed,

we have run the model with the same conditions as the ones described in section 3 of the original manuscript, but with fixed bathymetry and land-sea mask to the LGM, that is, without applying the algorithms. Figure A1 shows an overview of a comparison between both simulations. As an example of some variables, we plotted time series of a) AMOC at 26N and 1000 meters depth; b) sea-ice extent in the Arctic; c) global SST and; d) global SSS for the run with variable (red) and fixed (blue) bathymetry and land-sea mask. We observe differences in the behaviour of the variables, particularly from 14 kyrs BP onward, when the ice-sheet melting rate is high and the changes in the coastline are large. Therefore, there are substantial differences between both simulations. However, a detailed study would be needed to explain the effects of applying our methodology in terms of the physical mechanisms and the climate response. We are planning to face it in another paper. In our current manuscript, instead, we intend to present the technical problem and the way we propose to solve it as a "Development and technical paper".

[Figure]

*Figure A1: Time series of a) AMOC at 26N and 1000 meters depth; b) sea-ice extent in the Arctic; c) global SST and; d) global SSS for a simulation of the last deglaciation with variable (red) and fixed (blue) bathymetry and land-sea mask.*

- **The authors should also emphasize the potential limitations of this method in terms of simulating abrupt events during deglaciation due to many linear processes they used, both in time and space. I perfectly understand smoothing procedures the authors described to avoid crash of the model, but during deglaciation many non linear changes occurred for**

**instance MPW and more generally acceleration of melting rates described for instance in C. Waelbroeck et al., Quaternary Science Reviews 21, 295-305, 2002, for the last 30k. Therefore the authors should discuss in more details what is the compromise between avoiding crash and capturing real non linear events.**

We apply our methodology to MPIOM, the ocean component of the MPI-ESM. We are not computing a variable topography in response to the melting rates and the isostatic adjustments. Instead, our algorithms read the topography fields in high resolution and construct a usable bathymetry to run the ocean model in a coarse resolution. Thus, changes in topography due to the ice-sheet growth or decay and the isostatic adjustments of the bedrock are prescribed input data for our tool. In the experiment we present in section 3 of the manuscript, we use the ICE6-G reconstructions to construct the prescribed high-resolution topography. Changes in topography can also be solved by an ice-sheet model and a solid earth model coupled to the ESM, but these changes are computed neither by the ocean model nor by our algorithms. In that sense, the abrupt events and non-linear changes in the melting rates that took place during the deglaciation are not affected by our procedure. We agree with the reviewer, that the abruptness of some of the past changes is not captured by linearly interpolating between time slices 500 years apart. We did not produce these data, so we had to work with what was available. However, we should stress, that the PMIP deglacial simulation is not the goal, but only a simple test bed. Our ultimate goal is the fully coupled model with atmosphere, ocean, ice sheets and solid earth, which automatically generates higher resolution (in time) signals.

When applying our methodology in a transient simulation, the changes in bathymetry and land-sea mask are limited for the ocean model, but those limitations are not affecting the evolution of the bottom topography due to the ice-sheet retreat. The algorithms read the high resolution topography and allow only small changes when generating the coarse resolution bathymetry to run MPIOM. This fact can slow down the flooding and drying events of the shelves regarding the ocean domain. Therefore, due to the smoothing method, the propagation of the coastline is affected. In any case, if this is a problem for the solution, the algorithms can be applied more often (every year, for example). From the results shown in section 3 of the manuscript, we conclude that changing the bathymetry every 10 years during the last deglaciation is an optimal compromise for our model setup between both, model performance and computing time. In general, the stencil for adaptation could be widened, which would allow a faster flooding of e.g. the Hudson Bay. This might be necessary also when using a model with higher horizontal resolution.

We clarify this point in section *4 Remarks*:

"There are mainly three limitations in our technique. First, the fact that changes in depth and coastlines are limited can slow down the flooding and drying events of the shelves. However, it is important to note that changes in topography in response to the ice-sheet retreat and isostatic adjustments are solved neither by the ocean model MPIOM nor by our algorithms. Instead, the HR topography is prescribed to our tool or solved by the ice-sheet model. In this sense, the non-linear changes or abrupt events that occurred during the last deglaciation are not affected by our methodology. Still, if the timing of the flooding and drying events of the shelves is considered to be critical, the algorithms could be applied more often within the simulation (every year, for example). However, in MPI-ESM, changing the topography implies also changes in the river routing and the land mask for the atmospheric model. Therefore, there should be a compromise between the frequency that topography is being changed and the computational time. From our

results, we conclude that changing the bathymetry every 10 years during the last deglaciation is an optimal compromise between both, model performance and computing time. Another possibility would be to widen the stencil used for collecting water for new ocean points. This would allow a faster propagation of coastlines by more than one grid point per iteration. This might also turn out to be necessary when applying the tool to ocean configurations with higher horizontal resolution."

**2. The authors should also clarify the part of the paper that may be directly useful for the PMIP4 deglaciation community and those that have been developed specifically for MPI ESM.**

We include it in section *4 Remarks:*

"Second, this tool was originally written for the curvilinear orthogonal grid (GR) with two poles. Although we presented in this paper the results for the coarse resolution GR30, the tool can be also applied for the low resolution (GR15) configuration of MPIOM. Still, for the moment its usage is limited to GR grids. We are currently working on a new version to include the tripolar (TP) quasi-isotropic grid (Murray, 1996) among the applications. In general, the algorithms are easily adapted to any ocean model that meets the same requirements as MPIOM: Arakawa-C grid in the horizontal, z-grid in the vertical including partial bottom cells, free-surface and mass flux boundary conditions. However, there are some parameters inside the scripts that depend on the grid. They are the location of each pair of points in order to perform the checking steps described in Sect. 2.1 for correcting the bathymetric details."

**Whereas this paper is worth to be published in GMD, I have also minor comments that it would be important the authors answer to before publication.**

**Minor comments:**

**Abstract:**

**A1 What do the authors mean by conservation of mass and tracers at regional scale. It is a bit misleading in the abstract. I think the authors have in mind to keep regional conservation when changing spatial resolution. They should clarify this issue.**

If some correction is needed to globally conserve mass and tracers, it is enough to distribute homogeneously a single value around the globe. By conserving mass and tracers at a regional scale, we mean that changes in a single grid point are not propagated globally. In other words, we avoid propagating water properties over long distances by affecting only regionally the potential changes in a single grid point.

We clarify this point in the *Abstract*:

"The strategy applied is described in detail and the algorithms are tested in a long-term simulation demonstrating the reliable behaviour. Our approach guarantees the conservation of mass and tracers at global and regional scales, that is, changes in a single grid point are only propagated regionally."

**A2 The authors, first tackle a very general problem: the bathymetry adaptation when simulating the last deglaciation. How far the algorithm developed here, beyond grid specificity can be easily**

**adapted to other models. A sentence in the abstract should clarify this point.**

We add a sentence in the *Abstract*:

"For the first time, we present a tool allowing for an automatic computation of bathymetry and land-sea mask changes in the Max Planck Institute Earth System Model (MPI-ESM). The algorithms developed in this paper can easily be adapted to any free-surface ocean model that uses Arakawa-C grid in the horizontal and z-grid in the vertical including partial bottom cells. The strategy applied is described in detail and the algorithms are tested in a long-term simulation demonstrating the reliable behaviour."

**Introduction:**

**I1 The first sentence is very general and partially untrue because of some aspects of the unprecedented speed of ongoing climate change. The authors should remove or modify this sentence.**

The sentence is removed:

"During the last deglaciation, the Earth transitioned from the last glacial to the present interglacial climate, experiencing a series of abrupt changes on decadal to millennium timescales."

**I2 The authors should mention that major uncertainties remain on reconstruction of Antarctica at LGM. Indeed, NH ice sheet reconstructions are better constrained, whereas Antarctica ice sheet reconstruction has often been an adjustable parameter. Therefore, the authors should mention Antarctica reconstruction uncertainties at LGM both from data and models (G. Philippon, Earth and Planetary Science Letters 248 (2006) 750.)**

The quality of the reconstructions is not the point of our paper. Our algorithms are applied to the ocean model and they do not care about the prescribed topography. Therefore, the tool we are presenting is independent of the uncertainties on reconstructions. We are using ICE6-G in our transient simulation just as a test case. We could also use Tarasov or even the modelled topography from the coupled ice sheet solid earth model PISM/VILMA as it is planned for the future. Anyway, we mention it in *1 Introduction*:

"Differences in ocean bathymetry and land-sea mask between present-day conditions and 21 ka BP calculated from the ICE-6G_C ice-sheets reconstructions (Argus et al., 2014; Peltier et al., 2015) are plotted in Fig. 1. In general, the topography of the NH ice sheets does not vary substantially between different reconstructions whereas uncertainties show larger for Antarctica (Abe-Ouchi et al., 2015). Values up to 125 meters in ocean depth variations (Fig. 1a) are estimated, representing deepening of the ocean with time. The largest changes in the oceanic boundaries occurred in the northern hemisphere where the extensive areas covered by ice sheets during the LGM were flooded due to the ice melting (blue areas in Fig. 1b). It is important, therefore, to consider these changes when attempting to simulate the last deglaciation, for example by including a varying ocean surface area and volume."

The following citation is added to *References*:

"Abe-Ouchi, A., Saito, F., Kageyama, M., Braconnot, P., Harrison, S. P., Lambeck, K., Otto-Bliesner, B. L., Peltier, W. R., Tarasov, L., Peterschmitt, J.-Y., and Takahashi, K.: Ice-sheet configuration in the

CMIP5/PMIP3 Last Glacial Maximum experiments, Geosci. Model Dev., 8, 3621-3637, https://doi.org/10.5194/gmd-8-3621-2015, 2015."

**I3 The authors should also mention that there have been already many successful publications on glacial-interglacial simulations cycles from EMIC (A. Ganopolski et al., Nature 529, pages 200–203 (2016)) and from GCM (A. Abe-Ouchi et al., Nature 500, (2013)190. Moreover, the authors should better emphasize what in this context would be the added value of accounting for sea level rise.**

We add your suggestions to *1 Introduction*:

"Moreover, some research was carried out by using comprehensive climate and ice-sheet models (Abe-Ouchi et al, 2013) or climate models interactively coupled with a dynamic ice-sheet model for studying the last glacial-interglacial cycles (Bonelli et al., 2009; Heinemann et al., 2014; Ganopolski et al., 2016) and more specifically, the LGM (Ziemen et al., 2014). Still, in standard ESMs, land-sea mask is traditionally treated as fixed."

Later in the same paragraph:

"In the PMIP4 last deglaciation Core experiment design, the bathymetry and land-sea mask are considered boundary conditions that cannot evolve automatically in the model. Thus, the decision of how often to make manual updates was left to the expert (Ivanovic et al., 2016). However, by varying the bathymetry in small steps, the artificial signals produced by changes in the ocean configuration might be reduced yielding to a more realistic representation of the ocean circulation and its interaction with the other climate components during the last deglaciation.¨

The following citations are added to *References*:

"Abe-Ouchi, A., Saito, F., Kawamura, K., Raymo, M. E., Okuno, J. I., Takahashi, K., and Blatter, H.: Insolation-driven 100,000-year glacial cycles and hysteresis of ice-sheet volume, Nature, 500, 190-194, 2013.

Ganopolski, A., Winkelmann, R., Schellnhuber, H. J.: Critical insolation–CO2 relation for diagnosing past and future glacial inception. Nature, 529 (7585): 200 DOI:10.1038/nature16494, 2016."

**I4 Superimposed to the vertical resolution of MPIO, an important issue to be discussed is the choice of the initial horizontal resolution.**

We clarify this point in *1 Introduction*:

"In this paper, we use the MPIOM coarse resolution configuration with a curvilinear orthogonal grid (GR30) and two poles (Haak et al., 2003), over Greenland and Antarctica. We decide to use the coarse configuration to reduce the computational time, but the algorithms presented in this paper can easily be adapted to higher resolution grids. In the vertical, the model has 40 unevenly spaced levels, ranging from 15 meters near the surface to several hundred meters in the deep ocean."

**Methodology:**

**M1 It is not clear for me that accounting for only two big lakes (Caspian and Black Sea), the authors can capture abrupt climate changes occurring during deglaciation, as for instance the 8.2 ka event. Moreover, the evolution of Caspian and Black Sea associated to Eurasian ice-sheet melting and large modification of the catchment is not easy to be reconstructed and depicted. The authors should clarify more explicitly what is the limit of their method. Specifically, they should explain how they cope with river run-off and changes in catchment areas during deglaciation for these two epicontinental seas. These issues have been shown to have drastic consequences on atmosphere and ocean circulation (see for example R. Alkama et al., GRL 33 (21) 2006, R. Alkama et al., 2008, Climate Dynamics. 30 and M. Wary et al, J. Quaternary Sci. 32, 908–922, 2017).**

Actually, we are not accounting for lakes in order to capture the abrupt climate changes. Our algorithms are applied within the ocean model and therefore, they work on the ocean domain. In that sense, we are interested only in lakes that are connected to the ocean, that is the Black Sea. The Caspian Sea is, indeed, an exemption because it is not connected to the oceans. However, the Caspian Sea is much larger than the other minor lakes. We decided to include it to solve the SST there that might impact on the climate of Central Asia. Therefore, solving the SST of the Caspian Sea might be important for coupled climate models.

As it was mentioned before, we are presenting a tool that is independent of the uncertainties on the reconstructions. We are not solving the response of the topography to the ice-sheet melting and isostatic adjustments, but we are only prescribing them to our scripts. This is a problem to be accomplished by the ice sheet-solid earth models.

Finally, you are right that changes in catchment areas during deglaciation have drastic consequences on atmosphere and ocean circulation. But, this is a problem that is treated in the hydrologial discharge model (part of the land module) instead of the ocean as described by Riddick et al. (2018), as it is stated in the manuscript.

We clarify these issues at the beginning of section *2 Methodology*:

"Finally, we check for the presence of lakes in the GR30 bathymetry; the Caspian Sea and the Black Sea (under LGM condition, for example) are the only cases that are permitted. Because we are dealing with an ocean model, we are interested in lakes that are connected to the ocean, that is the Black Sea. However, we include the Caspian Sea in our calculations because of its potential impact on the climate of Central Asia. Solving the SST of the Caspian Sea, which is much larger than other minor lakes, might be important for coupled climate simulations. All other lakes need to be removed from the ocean domain either by connecting them to the open ocean or by considering them as land. The atmospheric model component allows accounting for lakes on land (only the thermal component). The adequate place to calculate water storage in lakes is the hydrological discharge model."

**M2 At the end of paragraph 2.3, in the spatial smoothing procedure for SSH, there are also changes in water mass reorganization that lead to spatial variations of the sea level rise during melting as shown for instance in Mitrovica (Nature 2001,...). Is this effect accounted for? If not, the authors should clarify the possible impact of this process.**

This issue is not specifically part of our algorithms but of the HR prescribed topography that enters to our scripts as input data. We assume that those effects are accounted for in the prescribed topography

which should already contain the gravitational adjustments. In the fully coupled simulation that we are planning to run, the effects you mention are solved by the ice sheet-solid earth component.

**Results:**

**R1 Whereas this paper is submitted for publication in GMD and devoted to technical and model development aspects, it is difficult to consider the validity of the process only analyzing the stability of the response without any information on the potential climate effect. Indeed, accounting for bathymetry with time steps of 10 years should allow the authors to capture the complex pattern of the deglaciation periods. Nevertheless, due to linear smoothing in time and space, it is unclear to me whether they really may capture abrupt events. This limitation should be discussed in more details.**

As it was discussed before, the aim of this paper is to present a tool that allows for automatic changes of bathymetry and land-sea mask in the ocean component of MPI-ESM. We are not attempting here to analyse the climate response to a changing topography. The way in which the inclusion of this tool affects on the deglaciation will be studied in the future. The transient simulation exposed in section 3 has the purpose of testing the algorithms in a long-term run. By testing the algorithm we mean the evaluation of the tool in terms of model stability and conservation of water properties as it is stated at the beginning of section *3 Transient simulation*:

"This section has the aim of testing the above-described tool in a long-term run with MPI-ESM. The purpose is not to analyse the climate response to a changing bathymetry and land-sea mask, this will be discussed in a consecutive paper. The aim of this experiment is evaluating the performance of the tool in terms of model stability and conservation of mass and tracers. This is a necessary step towards a fully coupled simulation."

Due to the changes in the model domain, the fields of SSH and tracers from the restart file are modified. It is therefore important to maintain the same amount of water and tracers inside the system. The choice of 10-years is not crucial for such an evaluation. Knowing that the algorithms guarantee the conservation of water properties, the tool can be applied more often if necessary. Beside this, we are discussing the abrupt flooding of the Hudson Bay in our algorithm.

**R2 Superimposed to ice sheet melting, a major component of the SLR is the ocean thermal expansion during deglaciation. Therefore it should produce a difference between SLR and cumulative fresh water input. In fig. 8, I suggest to plot, superimposed to the black and red curves, the component relative to the changes of the ocean volume associated with the thermal expansion during deglaciation.**

Thermal expansion is, indeed, not included in the model. MPIOM, as many other ocean models, uses the incompressibility assumption. As a consequence of this, tracers are conservative relative to volume and not relative to mass and the model conserves volume and not mass. Including the thermal expansion term in an ocean model as MPIOM is not consistent with the model physics because it would imply to give up the incompressibility assumption. In any case, the relative effect of thermal expansion on SSH is small compared with the signal due to the freshwater input. We include it in the model description in section *1 Introduction*:

"MPIOM is a free-surface ocean general circulation model with the hydrostatic and Boussinesq approximations and incompressibility assumption."

We also include a sentence in section *3 Transient simulation*:

"The difference between both time series was divided by the ocean area in order to obtain the errors in mean sea level (Fig. 8b). They are of the order of $1\times10^{-3}$ cm and within the computational accuracy. Therefore, the changes in ocean volume match the freshwater input indicating that water is being conserved. Note that MPIOM uses the incompressibility equations and therefore, the contribution of the thermal expansion on SSH is not being considered here. The year when the Black Sea is connected to the Mediterranean Sea, around 10.3 ka BP, is an exception for the conservation."

Finally, we do not see the advantage of showing a plot of the thermal expansion in the paper. In addition, changes in volume and mean salinity make it rather tedious to calculate. Traditionally, in ocean models with the incompressibility assumption, thermal expansion can be calculated by using the volume integral over the density, thus giving a mass. The difference of the calculated masses between 2 time slices, can be converted into a volume change (in a real ocean we can assume that volume changes and mass doesn't). Then, dividing the change in ocean volume by the ocean area yield the sea level change by thermal expansion (as was done e.g. in Mikolajewicz et al. 1990). However, here we have the problem that the volume within the ocean model is no longer constant, but the changes are substantial. Therefore, we would have to feed in further assumptions how the additional water entering the ocean should affect the reference mass used for calculation of thermal expansion. As changes in volume are quite large (more than 100 meters in sea level during our simulation) and much larger than the expected value of the thermal expansion (probably a few meters), we would expect quite some uncertainty in the estimation of thermal expansion because of the assumption how to deal with the extra water for the calculation of the reference value. This issue would require a discussion about circulation and climate changes in our deglacial simulation, which is not the topic of this paper.

**R3 is the model accounting for a possible ice shelf at the beginning of the deglaciation in the northern hemisphere?**

No, MPIOM does not include ice shelves. Therefore, the transient simulation we present does not account for ice shelves in any moment of the run. We include it in the model description in section *1 Introduction*:

"MPIOM includes an embedded dynamic/thermodynamic sea-ice model (Notz et al., 2013) with a viscous-plastic rheology following Hibler (1979). Sea-ice is swimming in the water. Ice shelves are not included. In this paper, we use the MPIOM coarse resolution configuration with a curvilinear orthogonal grid (GR30) and two poles (Haak et al., 2003), over Greenland and Antarctica."

**Remarks:**

**RM1 As the impact on climate due to change in bathymetry is not described in this paper, we can still have in mind many questions concerning the limits of this tool, when applied to non linear processes as those occurring during deglaciation. Indeed, the deglaciation is far to be a linear process. Major abrupt events (MWP and HE) occurred that are associated with large increase of fresh water inputs. It would be interesting that the authors discuss these potential limitations.**

This issue, as already explained before, is not a limitation of our tool but of the prescribed topography and freshwater forcing. We are not solving the topography response to abrupt events, actually. The changes in topography associated with the large increase of freshwater inputs should be included in the

forcing we are prescribing and our algorithms do not depend on it. This would be a task for either the reconstructions or the ice-sheet model, and not for the ocean model. In the fully coupled model under development this will be a very interesting aspect and the model should be suitable to cope with it.

**Final comment:**
**This study is interesting and novel. Moreover, it corresponds to an awaited development to better simulate the last transient deglaciation. Therefore when the authors will have answered the questions raised above, the manuscript will be worth to be published.**

---

## Referee Report (RR1)

Second review on the paper entitled "Interactive ocean bathymetry and coastlines for simulating the last deglaciation with the Max Planck Institute Earth System Model " by Virna Loana Meccia and Uwe Mikolajewicz.

The new manuscript is fine for me and may be published in GMD.

**I have now very minor comments**

First, I agree that this paper copes with an important issue: accounting for sea level rise during deglaciation. The automatic solutions provided in this study is a step forward for transient simulations of the deglaciation. The fact that the study well captured chronology of sea level during the last 30ka with abrupt phases and more linear behavior is still not very clear. The time step 500 years is a severe limitation. This is why I suggest that the authors more clearly contrast the strong points of their study which are indeed a serious added value and the limitations in terms of capturing the real chronology of the deglaciation at high temporal resolution (YD, HE, MWP).

My second comment concerns the responses to my initial questions at the first round of the revision. The authors answered satisfactorily to most of my queries and clarified the manuscript. Nevertheless, they often argued that they choose to submit their manuscript to GMD because it is a rather technical paper and scientific questions will be tackled in future publications. I only partially agree with that answer. On the one hand, this method will certainly allow to investigate new questions and indeed it is an efficient tool and an important prerequisite to provide more realistic deglaciation runs with AOGCM, On the other hand, the method has to be validated to prove that in real cases, all these developments are useful. That is why I consider that the authors have to better emphasize the interest of their method comparing in more details their results with transient simulation without changing sea level and continental distribution.

In any case, we have there a good piece of work and the paper can be published directly.

---

## Referee Report (RR2)

Comments on Revised manuscript GMD-2018-129

The revised manuscript is clear, well written, and the figures convincingly demonstrate the approach. The revised work does an excellent job of documenting the algorithm for interactively and automatically updating ocean bathymetry and coastlines for long-term transient earth system simulations.  It represents a monumental step forward for paleoclimate modeling and should be published with a few remaining minor tweaks.

Minor comments:
1) Line 20: "…brick stone …"  *Brick stone* does not convey a metaphorical meaning in English.  I suspect it is a direct translation of a more meaningful phrase or word in the native language.
2) Line 63: Liu et al. 2009 updated the ice sheet topography about every 1000 years, and the land-sea mask at longer and fewer irregular intervals over the deglacial period, not uniformly 500 years as written here.
3) Line 114: "Sea ice is swimming in the water." Remove line as use of "swimming" is incorrect here and the line is probably not necessary anyway.